# The transcribed pseudogene *RPSAP52* enhances the oncofetal HMGA2-IGF2BP2-RAS axis through LIN28B-dependent and independent *let-7* inhibition

Cristina Oliveira-Mateos[1], Anaís Sánchez-Castillo[1,2], Marta Soler [1], Aida Obiols-Guardia[1], David Piñeyro [1], Raquel Boque-Sastre [1,3], Maria E. Calleja-Cervantes[1], Manuel Castro de Moura[1], Anna Martínez-Cardús [1], Teresa Rubio[4], Joffrey Pelletier [4], Maria Martínez-Iniesta[5], David Herrero-Martín [6], Oscar M. Tirado [2,6], Antonio Gentilella [4,7], Alberto Villanueva[5], Manel Esteller [1,2,8,9,10], Lourdes Farré[5,11] & Sonia Guil [1,10]

One largely unknown question in cell biology is the discrimination between inconsequential and functional transcriptional events with relevant regulatory functions. Here, we find that the oncofetal *HMGA2* gene is aberrantly reexpressed in many tumor types together with its antisense transcribed pseudogene *RPSAP52*. *RPSAP52* is abundantly present in the cytoplasm, where it interacts with the RNA binding protein IGF2BP2/IMP2, facilitating its binding to mRNA targets, promoting their translation by mediating their recruitment on polysomes and enhancing proliferative and self-renewal pathways. Notably, downregulation of *RPSAP52* impairs the balance between the oncogene *LIN28B* and the tumor suppressor *let-7* family of miRNAs, inhibits cellular proliferation and migration in vitro and slows down tumor growth in vivo. In addition, high levels of *RPSAP52* in patient samples associate with a worse prognosis in sarcomas. Overall, we reveal the roles of a transcribed pseudogene that may display properties of an oncofetal master regulator in human cancers.

[1] Cancer Epigenetics and Biology Program (PEBC), Bellvitge Biomedical Research Institute (IDIBELL), L'Hospitalet de Llobregat, Barcelona, Catalonia, Spain. [2] Centro de Investigación Biomédica en Red de Cáncer (CIBERONC), Carlos III Institute of Health (ISCIII), Madrid, Spain. [3] Cardiff School of Biosciences, Cardiff University, Museum Avenue, Cardiff CF10 3AX Wales, UK. [4] Laboratory of Cancer Metabolism, ONCOBELL Program, Bellvitge Biomedical Research Institute (IDIBELL), L'Hospitalet de Llobregat, Barcelona, Catalonia, Spain. [5] Program Against Cancer Therapeutic Resistance (ProCURE), ICO, IDIBELL, L'Hospitalet de Llobregat, Barcelona, Catalonia, Spain. [6] Sarcoma Research Group, ONCOBELL Program, Bellvitge Biomedical Research Institute (IDIBELL), L'Hospitalet de Llobregat, Barcelona, Catalonia, Spain. [7] Department of Biochemistry and Physiology, Faculty of Pharmacy, University of Barcelona (UB), Barcelona, Catalonia, Spain. [8] Physiological Sciences Department, School of Medicine and Health Sciences, University of Barcelona (UB), Barcelona, Catalonia, Spain. [9] Institució Catalana de Recerca i Estudis Avançats (ICREA), Barcelona, Catalonia, Spain. [10] Josep Carreras Leukaemia Research Institute (IJC), Badalona, Barcelona, Catalonia, Spain. [11] Laboratory of Experimental Pathology (LAPEX), Gonçalo Moniz Research Center, Oswaldo Cruz Foundation (CPQGM/FIOCRUZ), Salvador, Bahia, Brazil. Correspondence and requests for materials should be addressed to L.F. (email: mfarre@idibell.cat) or to S.G. (email: sguil@carrerasresearch.org)

The largest part of the mammalian genome is transcribed into RNA species with little or no coding potential, known as noncoding RNAs (ncRNAs)[1]. Although their biological roles are still largely unknown, a growing number of long noncoding RNAs (lncRNAs, a label arbitrarily assigned to transcripts longer than 200 nucleotides) display regulatory properties by acting at all levels in gene expression control (from epigenetic modifications and chromatin dynamics to the control of post-transcriptional messenger RNA stability and translation)[2,3]. In some cases, their key functions in normal homeostasis and development links the dysregulation of their expression with causal roles in cancer[4], and there are instances of lncRNAs involved in each of the cancer hallmarks, including sustained proliferative signaling and growth (e.g., ANRIL[5], lincRNA-p21[6], MEG3[7]), invasion and metastasis (e.g., HULC[8], MALAT1[9], HOTAIR[10]), resistance to cell death (e.g., PCGEM1[11]), and replicative immortality (e.g., TERC[12], TERRA[13]). Mechanisms of action include the interaction with other nucleic acids and/or protein factors, which confers the ability to function as scaffolds, guides, decoys, or allosteric regulators of several nuclear or cytoplasmic processes[14,15]. In a growing number of examples, their roles intertwine with that of the better-studied miRNAs[16], either by cooperating in their function[17] or by impairment of the miRNA-mediated regulation[18]. The latest annotation in GENCODE estimates that up to 16,000 genes in the human genome correspond to lncRNAs, and a similar number is given to pseudogenes (https://www.gencodegenes.org/stats/current.html#). Although some pseudogenes do code for proteins, the majority are thought to be lncRNAs owing to the accumulation of mutations in the definition of the open reading frames, and as such their biological functions include the ability to regulate gene expression similarly to lncRNAs[19], and are thereby also involved in growth-regulatory roles in cancer[20].

RPSAP52 is a pseudogene-transcribed RNA that runs antisense to the oncofetal gene HMGA2, a transcriptional co-regulator that is expressed at high levels during embryonic development, silenced in virtually all adult tissues and re-expressed in several human cancers, where its levels are generally associated with the presence of metastases and poor prognosis[21,22]. Our previous results indicate that RPSAP52 positively regulates HMGA2 expression through the formation of an R loop structure[23]. Herein we further study the role of this transcribed pseudogene in breast and sarcoma tumors, and uncover its role as a pro-growth factor through the regulation of the IGF2BP2/IGF1R/RAS axis and the balance between LIN28B and let-7 levels.

## Results

### RPSAP52 impacts on IGF2BP2 and let-7 in breast cancer cells.

We have previously uncovered the positive impact of the expression of the pseudogene RPSAP52 on its sense, protein-coding gene HMGA2 (Fig. 1a)[23]. Both genes are generally expressed at low levels in differentiated normal tissues and overexpressed in a number of human cancers, including breast cancer, concomitant with a hypomethylation of the associated CpG island (Fig. 1b). In breast cancer patients, a positive correlation between the expression of both genes is observed (Fig. 1c), as is also seen in the NCI60 panel of cell lines (Supplementary Fig. 1a). Other studies have reported that high HMGA2 expression predicts poor outcome in breast cancer patients[25]. Since our observations indicate that knockdown of RPSAP52 results in a reduction in HMGA2 expression[23], we decided to look further into the molecular mechanism of RPSAP52-mediated regulation of the locus. A panel of breast cancer cell lines was used to confirm the presence of RPSAP52 transcript by semi-quantitative PCR (Fig. 1d). Surprisingly, most cell lines expressed the annotated RPSAP52 transcript

(Refseq NR_026825.2) together with an additional species that corresponds to the inclusion of a 104-nucleotides-long internal exon (Fig. 1d and Supplementary Fig. 1b). Evidence as to the presence of this alternative exon in the spliced transcript can also be found in the MiTranscriptome database[24], which catalogs long polyadenylated RNA transcripts (www.mitranscriptome.org, with reference G018828|T081486). The quantitative measurement of expression levels indicates that HMGA2 mRNA and the two isoforms of RPSAP52 are 2-3 orders of magnitude overexpressed when there is hypomethylation of the promoter-associated CpG island, as shown with Illumina's HumanMethylation450 BeadChip analysis (Fig. 1e) and was confirmed by bisulfite sequencing at the nucleotide level (Supplementary Fig. 1c). Altogether, these observations confirm the coordinate expression of both genes and their silencing in hypermethylated conditions. RPSAP52 is annotated as a noncoding RNA in Refseq, but is labeled as coding in some coding potential calculator tools. Pseudogenes are more likely to give false positive results in programs such as PhyloCSF (since they are similar to their parental protein-coding, and PhyloCSF evaluates conservation to predict coding capacity). We thus conducted in vitro transcription/translation assays, which confirmed the absence of RPSAP52 coding potential (Supplementary Fig. 2a). However, analysis of RNA presence along sucrose gradients from MCF10A cells showed the presence of RPSAP52 transcripts in polysomal fractions, indicating a role in translation. Interestingly, a strong correlation in co-sedimentation of HMGA2 mRNA and RPSAP52 transcript was observed (Supplementary Fig. 2b–e). Indeed, further characterization of RPSAP52 transcripts showed that they are enriched in the cytoplasm (Fig. 1f) and polyadenylated (Fig. 1g), suggesting additional roles besides the ability to regulate HMGA2 transcription in the nucleus. In order to identify protein partners of RPSAP52 that could help characterize its activity, we performed RNA pull-down assays combined with mass spectrometry (MS). In vitro synthesized full-length RPSAP52 RNA was incubated in the presence of MCF10A extracts and the retrieved proteins were analyzed by SDS-PAGE. As shown in Fig. 2a, a protein band of ~70 kDa is specifically pulled-down by RPSAP52 RNA, but not by its antisense sequence or another unrelated RNA. This band was characterized by MS, which identified two proteins within the isolated fragment: the insulin-like growth factor 2 mRNA-binding protein 2 (IGF2BP2), also known as IMP2 (from which seven peptides were identified), and the heterogeneous nuclear ribonucleoprotein Q (HNRNPQ), also known as SYNCRIP (identified with six peptides) (Supplementary Fig. 3a). IGF2BP2, together with IGF2BP1 and IGF2BP3 partakes of a family of RNA binding proteins that have been implicated in post-transcriptional control, including the regulation of mRNA localization, stability, and translation[26]. Similar to HMGA2, although the expression of IGF2BPs is normally restricted to embryonic stages, they are re-expressed upon malignant transformation, playing roles in the maintenance of cancer stem cells and the promotion of tumor growth[27]. Western blot with specific antibodies confirmed that both IGF2BP2 and HNRNPQ are enriched in the RPSAP52 pull-down, and analysis of RPSAP52 truncates indicate that the two isoforms are able to bind to these two factors (Fig. 2b). In accordance with the pull-down results, the previously reported consensus binding site for IGF2BP2, the CAUH (H = A, C, U) motif[28], is abundant along the two constitutive RPSAP52 exons, but absent on the alternative exon (Supplementary Fig. 3b), suggesting that the alternative splicing event does not impact on the affinity of the binding.

IGF2BP2 is a direct transcriptional target of HMGA2[29] and both proteins partake of a pro-proliferative axis[30,31] that interweaves with the function of let-7 family of miRNAs. Both HMGA2 and IGF2BP2 mRNAs are direct targets of let-7, but IGF2BP proteins have been suggested to modulate let-7 action via

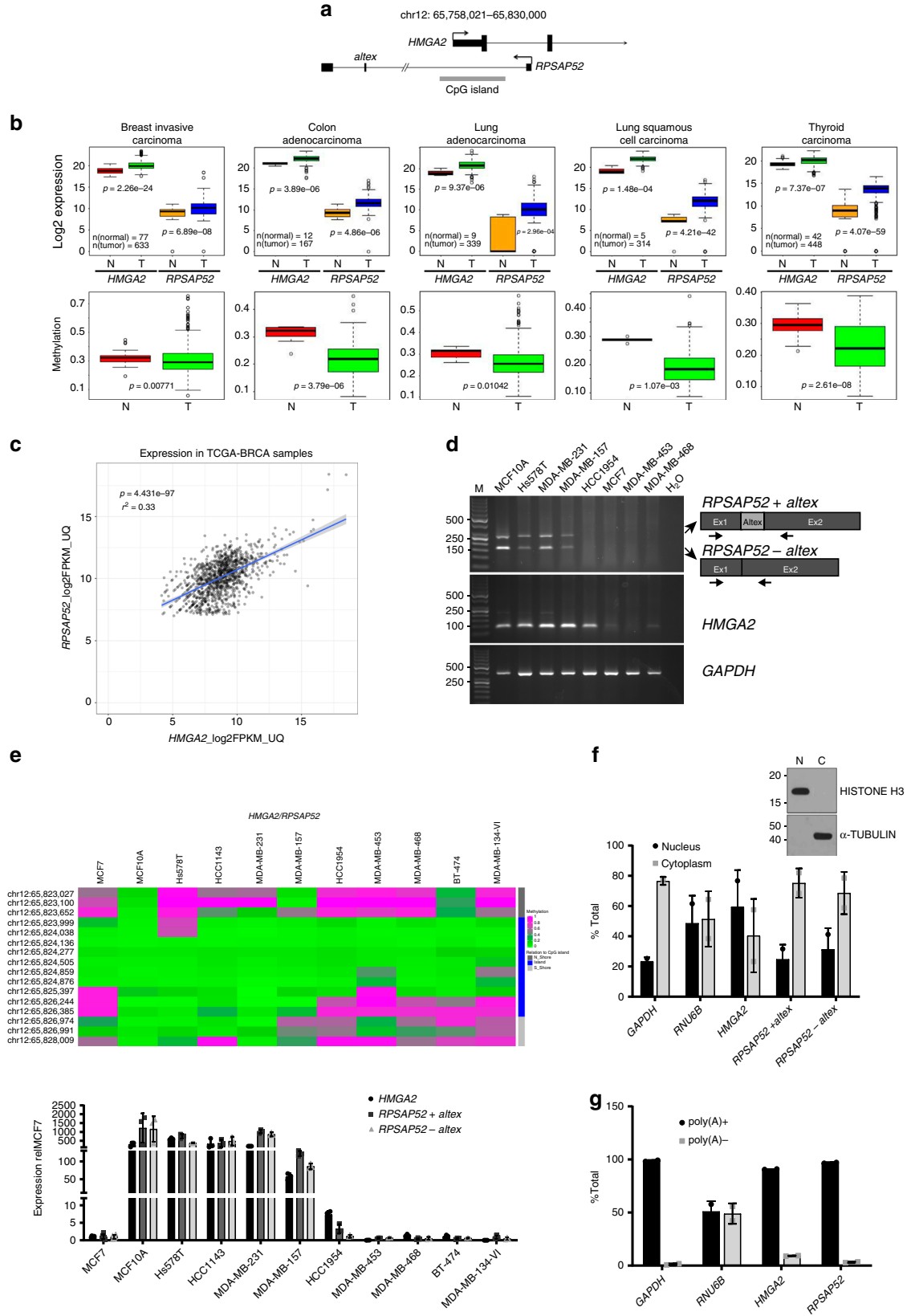

the formation of cytoplasmic mRNPs that would protect certain mRNAs from *let-7* binding and repression[32,33]. The most abundantly expressed members of the family are *let-7a/b/e* in MCF10A cells, and *let-7a/d/f/g/i* in Hs578T cells (Supplementary Fig. 3c). Interestingly, the levels of the mature form of these

miRNAs were upregulated in *RPSAP52*-depleted cells, both in MCF10A and Hs578T clones stably expressing shRNAs (Fig. 2c and Supplementary Fig. 3g) and in cells transiently expressing three different locked nucleic acid (LNA)-based antisense oligonucleotides (ASOs) gapmers (Fig. 2e). *RPSAP52* has a region

**Fig. 1** Characterization of *RPSAP52* expression in breast cancer. **a** Intronic/exonic organization of sense/antisense transcripts in *HMGA2 locus*. Coordinates are referred to the UCSC Genome Browser (GRCh38/hg38 release). Only the 5′ regions of *HMGA2* transcripts are included. **b** Box plot representations show both *HMGA2* and *RPSAP52* transcripts are overexpressed in a variety of human cancers compared with normal controls (TCGA dataset) (upper panels), concomitant with hypomethylation of the associated CpG island (lower panels). On each box plot, the central mark indicates the median, and the bottom and top edges indicate the interquartile range (IQR). The box plot whiskers represent either 1.5 times the IQR or the maximum/minimum data point if they are within 1.5 times the IQR. *P*-values are according to the two-tailed Wilcoxon signed-rank test. **c** Pearson correlation between *HMGA2* and *RPSAP52* transcripts in breast cancer primary tumors (all stages included). Normalized values of RNA-seq data from TCGA are represented. **d** Semi-quantitative RT-PCR to detect the expression of *HMGA2* and *RPSAP52* in a panel of breast cancer cell lines. Detection of *RPSAP52* transcripts was done with primers that detect both isoforms. **e** Upper panel: heatmap representation of the DNA methylation profile for the CpG island-containing promoter at the *HMGA2 locus*, as analyzed with the 450K DNA methylation microarray. Single CpG methylation levels are shown. Green, unmethylated; magenta, methylated. Data from 11 breast cancer cell lines are shown. Lower panel: the expression levels of both *RPSAP52* (including or excluding the alternative exon) and *HMGA2* were analyzed by RT-qPCR and represented relative to MCF7 cell line. Graphs represent the means of three replicates from different RNA extractions ±SD. **f** Nuclear/cytoplasmic fractionation of MCF10A cells, analyzed by RT-qPCR and western blot to assess fraction purity. Graphs represent the mean ±SD of two replicates of fractionation. **g** Poly(A)+/poly(A)− partition of total RNA from MCF10A cells and analysis by RT-qPCR. Primers that detect the *RPSAP52* +*altex* isoform were used. Graphs represent the mean ±SD of two replicates of poly(A) selection. Source data are provided as a Source Data file

of homology with other *RPSA* pseudogenes, but none was affected by our depletion strategy (Supplementary Fig. 3d). Also, given the possibility that some pseudogenes regulate parental gene expression, we analyzed the levels of RPSA protein in the *RPSAP52*-depleted cells, but no quantitative change was found (Supplementary Fig. 3e). Of note, gapmer-mediated depletion of *HMGA2* increased both *RPSAP52* isoforms and resulted in a decrease in *let-7* levels, suggesting that the negative regulation exerted by *RPSAP52* on the miRNAs is not through HMGA2 pathway (Fig. 2e). *Let-7* regulates *IGF2BP2* mRNA and other members of IGF1 signaling pathway, among others *IGF1R* and *RAS*. Down-regulation of *RPSAP52* with shRNAs or gapmers reduces the amount of these proteins, in accordance with the increased *let-7* levels (Fig. 2d, f and Supplementary Fig. 3g, h). LIN28A and LIN28B are the main negative regulators of *let-7* biogenesis, through direct binding to either *pre-let-7* and/or *pri-let-7*[34,35], but often only one of the two proteins is found expressed in human cancer cell lines[36]. We could not detect LIN28B protein expression in MCF10A cells, and LIN28A was not altered upon *RPSAP52* knockdown (Supplementary Fig. 3f), suggesting that changes in *let-7* in MCF10A cells were not consequences of impaired biogenesis at the level of regulation by LIN28. However, *RPSAP52*-mediated regulation of IGF2BP2 protein levels is reverted by overexpression of LIN28B, indicating a convergence on the same regulatory network (Fig. 2g).

Next, the phenotypic impact of the altered control of the IGF2BP2/IGF1R/RAS pathway by *RPSAP52* was tested both in vitro and in vivo.

**RPSAP52 has oncogenic-like features in vitro and in vivo.** Upon *RPSAP52* knockdown, all three breast cell lines tested (the non-transformed MCF10A and the tumorigenic Hs578T and HCC1143 cells) proved to be significantly less proliferative in the sulforho-damine B (SRB) assay (Fig. 3a), and had a significantly lower percentage colony formation density than control cells (Fig. 3b). Interestingly, *RPSAP52* depletion was also associated with a decreased migration potential (Fig. 3c). High levels of *let-7* miRNAs often correlate with a lower capacity for self-renewal and plur-ipotency. Given the observed reduction in proliferation and migration following *RPSAP52* knockdown, we next assessed the levels of markers of cell stemness (Fig. 3d). NANOG and OCT4 protein levels were decreased in *RPSAP52*-depleted cells, suggesting this lncRNA promotes features of cancer stem cells. This was fur-ther confirmed in soft-agar colony formation experiments, in which the measure of the anchorage-independent growth of the cells showed a significant decrease upon depletion of *RPSAP52* (Fig. 3e). For the in vivo approach, we next used tumor formation assays in

nude mice. MCF10A and Hs578T cells stably expressing either scrambled shRNAs or shRNAs against *RPSAP52* were sub-cutaneously injected into mice, and the tumor formation and volume was monitored. Tumors originating from *RPSAP52* knockdown cells had a significantly lower volume and weight at end point than control tumors, both for the non-tumorigenic and the tumorigenic cells (Fig. 3f, g). Importantly, the amount of RAS and IGF2BP2 proteins were markedly reduced in the excised tumors at end point, indicating that the in vitro findings were maintained in the in vivo context (Fig. 3f, g).

**RPSAP52 regulates IGF2BP2/LIN28B/let-7 axis in sarcoma.** We next attempted to determine whether *RPSAP52*-mediated reg-ulation of proliferative pathways occurred in other cancer types. The analysis of the collection of human cancers available from The Cancer Genome Atlas (TCGA) indicates *HMGA2* and *RPSAP52* expression is specially increased in adrenocortical car-cinoma, mesothelioma and in the sarcoma samples available (as measured by Z-score, Supplementary Fig. 4a). TCGA RNA expression and DNA methylation data showed that *RPSAP52* promoter hypermethylation was associated with transcript downregulation across sarcoma samples (Fig. 4a, upper panel; Pearson correlation, $r^2 = 0.264$, *P*-value = 4.764e–10). Of note, *HMGA2* expression in the same samples shows a poorer corre-lation (Supplementary Fig. 4b, $r^2 = 0.131$, *P*-value = 2.582e–05). Both genes maintain a positive expression correlation (Fig. 4a, lower panel) and a difference of ~1–2 orders of magnitude in their relative expression (Supplementary Fig. 4c). This is in agreement with absolute quantification of *RPSAP52* and *HMGA2* transcripts in MCF10A and A673 cell lines, in which *HMGA2* mRNA is 1–2 orders of magnitude at higher levels (Supplemen-tary Fig. 4d). Since the HMGA2-IGF2BP2-RAS pathway has been previously involved in the pathogenesis of embryonic rhabdo-myosarcoma[31], we then assessed *HMGA2* and *RPSAP52* expres-sion in a panel of cell lines derived from rhabdomyosarcoma and also Ewing's sarcoma (Fig. 4b). Both *RPSAP52* isoforms were abundantly expressed in most rhabdomyosarcoma cell lines, with just one order of magnitude higher *HMGA2* expression in Rh28, Rh41, or CW9019 cells. In Ewing's sarcoma, *RPSAP52* was gen-erally lowly expressed with the exception of A673 cell line. We thus focused on A673 cells to further characterize the molecular function of this lncRNA. As seen in MCF10A cells, RNA pull-downs confirmed the ability of *RPSAP52* to interact with IGF2BP2 and SYNCRIP/HNRNPQ (Fig. 4c). Importantly, stable clones expressing two different shRNAs against both *RPSAP52* isoforms resulted in a strong increase in *let-7* family members (Fig. 4d, upper panel), even when *HMGA2* levels were only

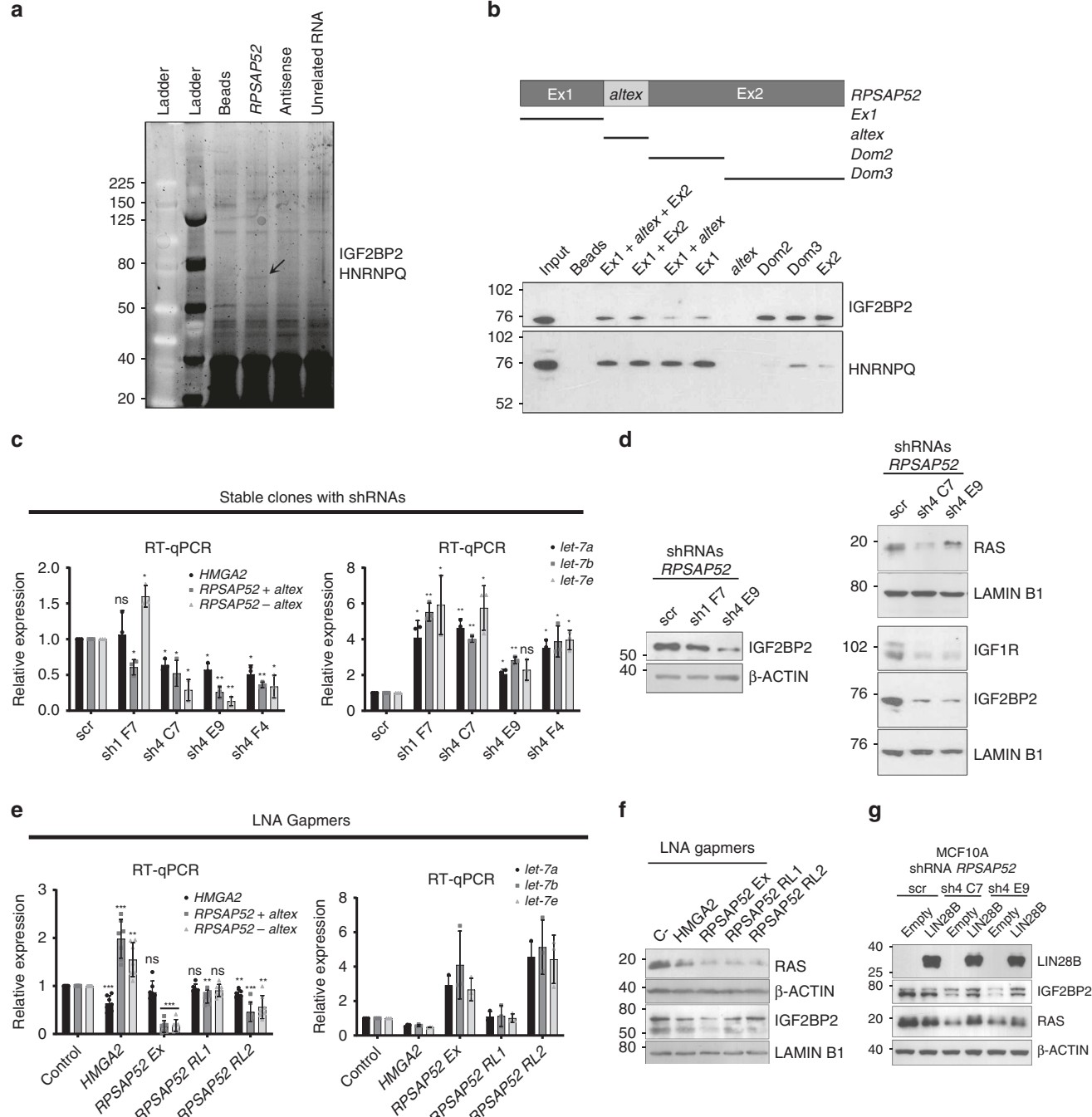

**Fig. 2** *RPSAP52* interacts with IGF2BP2 and HNRNPQ and influences proliferative pathways in MCF10A cells. **a** RNA pull-down assay to detect *RPSAP52*-associated proteins. In vitro synthesized full-length *RPSAP52* transcript (including the alternative exon) or control sequences (the antisense transcript and the unrelated Uc.160+ RNA) were tested. The proteins retrieved were analyzed by SDS-PAGE and the band of ~70 kDa indicated by the arrow was identified by MS as containing IGF2BP2 and HNRNPQ. **b** Western blot showing the association between *RPSAP52* RNA and IGF2BP2 and HNRNPQ proteins. Different truncated fragments of *RPSAP52* RNA (as shown in the upper diagram) were incubated in the presence of MCF10A total protein extracts and the pulled-down material was subject to western blot with specific antibodies. Total extract from the MCF10A cell lines was used as input control, and a reaction without RNA (beads) as negative control. **c** Stable knockdown of *RPSAP52* results in upregulation of *let-7* family of miRNAs. Total RNA from MCF10A clones constitutively expressing two different shRNAs against *RPSAP52* (sh1 or sh4) was analyzed by RT-qPCR to assess *HMGA2* mRNA, *RPSAP52* transcripts and *let-7* miRNAs levels. Graphs represent the mean ±SD of three independent RNA extractions. Two-tailed student *t*-test were used (\**P* < 0.05, \*\**P* < 0.01, ns = not significant). **d** Western blot to analyze IGF2BP2, IGF1R, and RAS protein levels upon stable knockdown of *RPSAP52* transcripts. **e** Transient transfection of MCF10A cells with locked nucleic acid (LNA)-based antisense oligonucleotides (ASOs) gapmers targeting *HMGA2* mRNA (HMGA2), exon1 (RPSAP52 Ex) or the first intron (RPSAP52 RL1 and RL2) of *RPSAP52* transcript. Expression levels of *HMGA2*, *RPSAP52* and *let-7* were measured by RT-qPCR. Graphs represent the mean ±SD of seven independent replicates (for *HMGA2* and *RPSAP52*) or three replicates (for *let-7*). Two-tailed student *t*-test were used (\**P* < 0.05, \*\**P* < 0.01, \*\*\**P* < 0.001). **f** Western blot to analyze IGF2BP2 and RAS protein levels upon transient knockdown of *RPSAP52* transcripts. **g** Western blot to analyze IGF2BP2 and RAS protein levels upon transient overexpression of LIN28B protein in the background of *RPSAP52* depletion. Source data are provided as a Source Data file

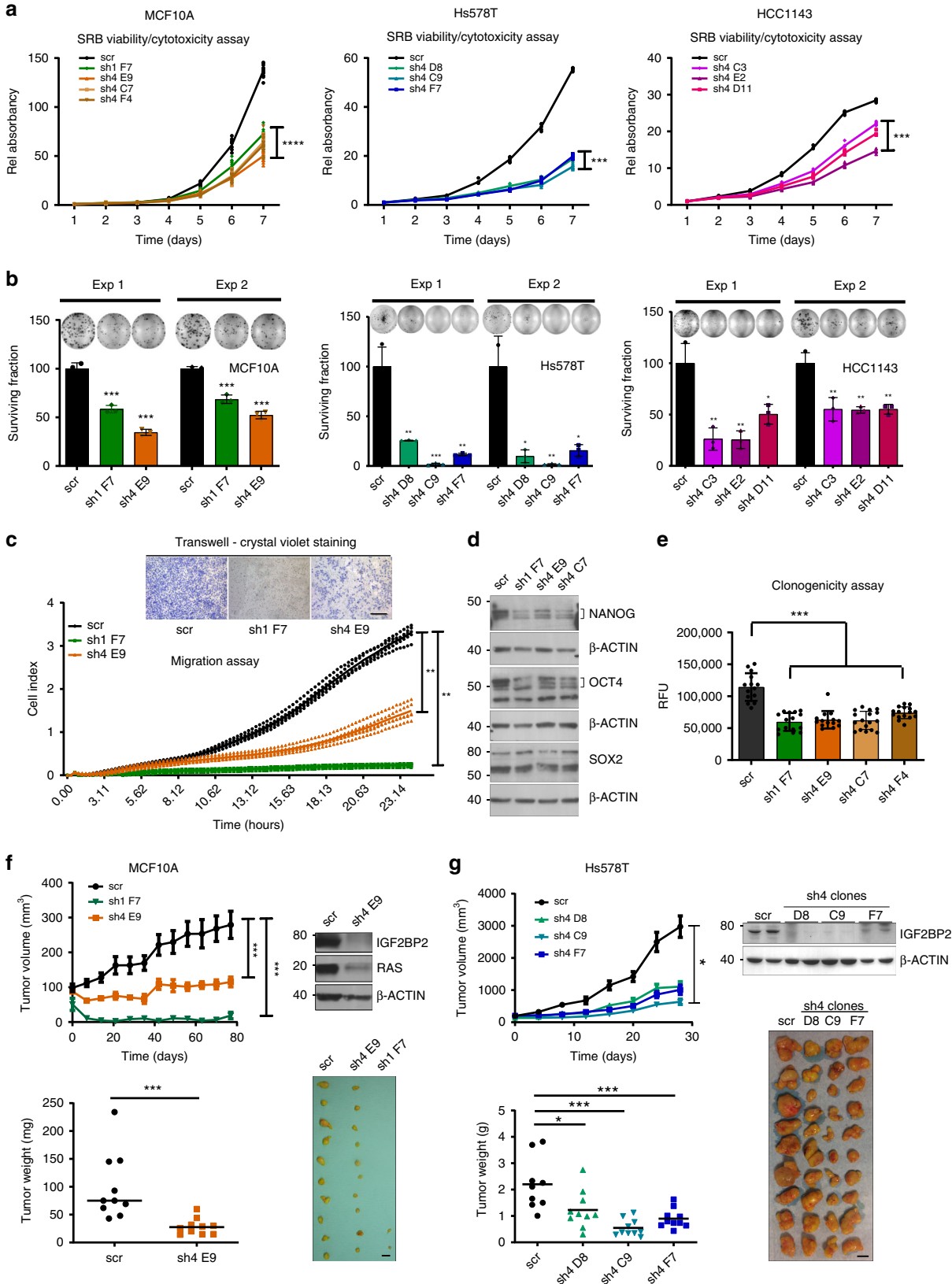

moderately reduced (Fig. 4d, lower panel). In this case, *RPSAP52* knockdown did not correlate with IGF2BP2 decrease, but with a marked reduction in LIN28B protein levels, which in contrast to breast cell lines, is abundantly expressed in A673 cells (Fig. 4e and Supplementary Fig. 4e). Also, while RAS levels were only partially

reduced, downstream signaling was impaired, as observed by the decrease in p-ERK levels (Fig. 4e). In an in vivo setting, and similarly to the observations in breast cell lines, this results in a marked reduction in tumor formation when mice are subcutaneously injected with *RPSAP52*-depleted A673 cells (Fig. 4f).

**Fig. 3** *RPSAP52* displays oncogenic features in breast cancer cells. **a** Viability/cytotoxicity assays in MCF10A, Hs578T and HCC1143 clones. The experiment was performed three times and one representative graph is shown for each cell line. Values are mean ±SD of $n \geq 6$ measurements. One-way ANOVA was used (***$P < 0.001$, ****$P < 0.0001$). **b** Effect of *RPSAP52* silencing on colony formation ability. Representative plates are shown. Colonies were counted from three replicate plates and two independent experiments. Values are mean ±SD. Two-tailed unpaired *t*-test were used (*$P < 0.05$, **$P < 0.01$, ***$P < 0.001$, ****$P < 0.0001$). **c** Migration capacity of *RPSAP52*-depleted clones was monitored over 24 h ($n = 5$ replicates per condition), with higher cell index indicating higher migration. Values are mean ±SD. A two-tailed Mann–Whitney U test of data at end point was used (**$P < 0.01$). Inset: migration was also assessed with transwells. Scale bar = 100 μm. **d** Western blot analysis of NANOG, OCT4, and SOX2 in *RPSAP52*-depleted cells. **e** The clonogenic ability was assessed with at least $n = 12$ replicates per condition. Values are mean ±SD. A two-tailed Mann–Whitney U test was used (***$P < 0.001$). **f** Growth-inhibitory effect of *RPSAP52* knockdown in MCF10A mice xenografts. Upper graph: tumor volume ($n = 10$) was monitored over time. Mean values are shown ±SEM. Lower graph: tumors were excised and weighed at 77 days (***$P < 0.001$, two-tailed Mann–Whitney U test). Western blot was carried out from sh4 tumors since no material could be recovered from sh1 tumors, and the levels of RAS and IGF2BP2 proteins were analyzed. The photograph shows the relative size of all tumors extracted. Scale bar = 10 mm. **g** Growth-inhibitory effect of *RPSAP52* knockdown in Hs578T mice xenografts. Upper graph: tumor volume ($n = 9$ for scr and $n = 10$ for sh4 clones) was monitored over time. Mean values are shown ±SEM. Lower graph: tumors were excised and weighed at 28 days (*$P < 0.05$, ***$P < 0.001$, two-tailed Mann–Whitney U test). Western blot was carried out from tumors at end point and the levels of IGF2BP2 protein were analyzed. The photograph shows the relative size of all tumors extracted. Scale bar = 10 mm. Source data are provided as a Source Data file

Interestingly, LIN28B protein reduction is only partially explained by a decrease in *LIN28B* mRNA levels (Supplementary Fig. 4f). This discrepancy, together with the interaction detected between *RPSAP52* and IGF2BP2, and the presence of *RPSAP52* along sucrose gradient's heavy fractions, which correspond to translating poly-ribosomes (see text above and Supplementary Fig. 2d), prompted us to investigate the possibility that LIN28B levels were regulated by the lncRNA at the translational level.

**RPSAP52 modulates IGF2BP2 binding to its mRNA targets.** IGF2BP2 is a mRNA stability and translational regulator with some well-described targets, such as IGF2[37], NRAS[31], or HMGA1[38]. The closely related IGF2BP1 protein has been shown to interact with *LIN28B* mRNA and increase LIN28B protein levels in ES-2 cells[32]. In order to assay the interaction of IGF2BP2 with *LIN28B* mRNA in A673 cells, we carried out protein immuno-precipitation followed by RT-qPCR of the pulled-down RNA. We could confirm the interaction of IGF2BP2 with both *RPSAP52* isoforms, with *IGF2BP2* and *NRAS* mRNA, and importantly, with *LIN28B* mRNA. Of note, even though the *RPSAP52* isoform lacking the alternative exon is more abundant in A673 cells, both transcripts were recovered in comparable amounts in IGF2BP2 immunoprecipitate, with a ~10-fold higher affinity of IGF2BP2 for *RPSAP52 + altex* RNA (Fig. 5a, see RT-qPCR). Further, LIN28B protein was not co-immunoprecipitated with IGF2BP2 protein (Fig. 5b), suggesting its putative regulation by IGF2BP2 is at the level of transcript. We next wanted to test the possibility that this binding is regulated by *RPSAP52* presence. Interestingly, whereas binding to *IGF1R* and *IGF2BP2* mRNAs was not altered, binding of IGF2BP2 to *LIN28B* mRNA was reduced upon stable knockdown of *RPSAP52* (Fig. 5c). This suggests that this lncRNA might regulate *LIN28B* post-transcriptionally through modulation of IGF2BP2 function. In view of this, we decided to characterize in a transcriptome-wide manner the IGF2BP2-RNA interactions with individual nucleotide resolution (iCLIP-seq) under control or *RPSAP52*-knockdown conditions. We identified 290,060 and 131,729 iCLIP-tags in control and *RPSAP52*-depleted A673 cells, corresponding to 3075 and 1639 peak regions, respectively (Supplementary Fig. 5a, b). As has been shown before, IGF2BP2 iCLIP tags were enriched in 3′UTRs[28,33], with ~60% of the iCLIP peaks falling within 3′UTRs in control cells. Remarkably, knockdown of *RPSAP52* resulted in a specific decrease in the number of 3′UTR peaks revealed by iCLIP and an increase in intronic regions (Fig. 5d). Motif enrichment analysis around the crosslinking-induced truncation sites (CITS) indicate that the previously described CAUH (H = A, C, U) consensus binding site[28] also ranks high in

our iCLIP experiments, but is more enriched in the control samples (Supplementary Fig. 5c, d). We found 1775 and 810 peaks with the CAUH motif in the control and depleted sample, respectively, representing a statistically significant difference in occurrence (Fisher's exact test *P*-value = 5.286e–08). In addition, the number of peaks with more than one CAUH motif was higher in the control cells (average number of motifs was 1.81 for control and 1.64 for depleted cells; Mann–Whitney U test, *P*-value = 1.786e–05). The full list of statistically significant iCLIP-seq peaks and CITS can be found in Supplementary Data 1. Differences in motif binding and the reduction in 3′UTR recognition results in a shortlist of 34 transcripts with differential IGF2BP2 iCLIP counts along their 3′UTR (Fig. 5e). GO enrichment analysis shows these genes belong to categories that may relate to IGF2BP2 involvement in cancer invasion and metastasis, including cell-substrate adhesion, spreading, and wound healing, as well as the canonical function for IGF2BP2 pathway, cellular glucose homeostasis (Fig. 5e). Of note, previous CLIP experiments for IGF2BPs in pluripotent stem cells have revealed that cell adhesion is also the most significant GO category for CLIP-enriched 3′UTRs for IGF2BP1[39]. This suggests that the levels of *RPSAP52* have a dramatic impact on IGF2BP2 global role. In addition, top ten GO categories for genes with significant iCLIP peaks present on their 3′UTRs in the control sample correspond to signaling pathways and cell cycle progression, whereas none of these categories are enriched in the *RPSAP52*-depleted sample (Fig. 5f).

Previous CLIP-seq studies with IGF2BP2 had revealed binding sites on the 3′UTR of *LIN28B* mRNA in HEK293T cells[28], and we detected similar sites in our experimental setting and a tendency to decrease upon *RPSAP52* depletion, although without any statistical power (Fig. 5g). For other validated IGF2BP2 targets, such as *HMGA2*, we also detected abundant iCLIP signal corresponding to direct binding of IGF2BP2 to its 3′UTR. In this case, binding is dramatically lost upon *RPSAP52* knockdown (Fig. 5h), and a corresponding decrease in HMGA2 protein level is observed (Supplementary Fig. 5e). This is not a general phenomenon for all IGF2BP2 targets, since other well-characterized mRNA partners, such as *HMGA1*, *NRAS*, and *IGF1R* maintain comparable iCLIP signals in both control and *RPSAP52*-depleted conditions (Supplementary Fig. 5f). Our results thus suggest a specific loss of IGF2BP2 affinity for particular mRNA targets. Several of the best characterized IGF2BP2 targets are regulated by *let-7* (e.g., RAS, HMGA2…) but, interestingly, we could not find a differential presence of *let-7* miRNA recognition motifs along the 3′UTRs of the immunoprecipitated mRNAs in control or depleted samples (Fisher's exact test *P*-value = 0.8974). Also,

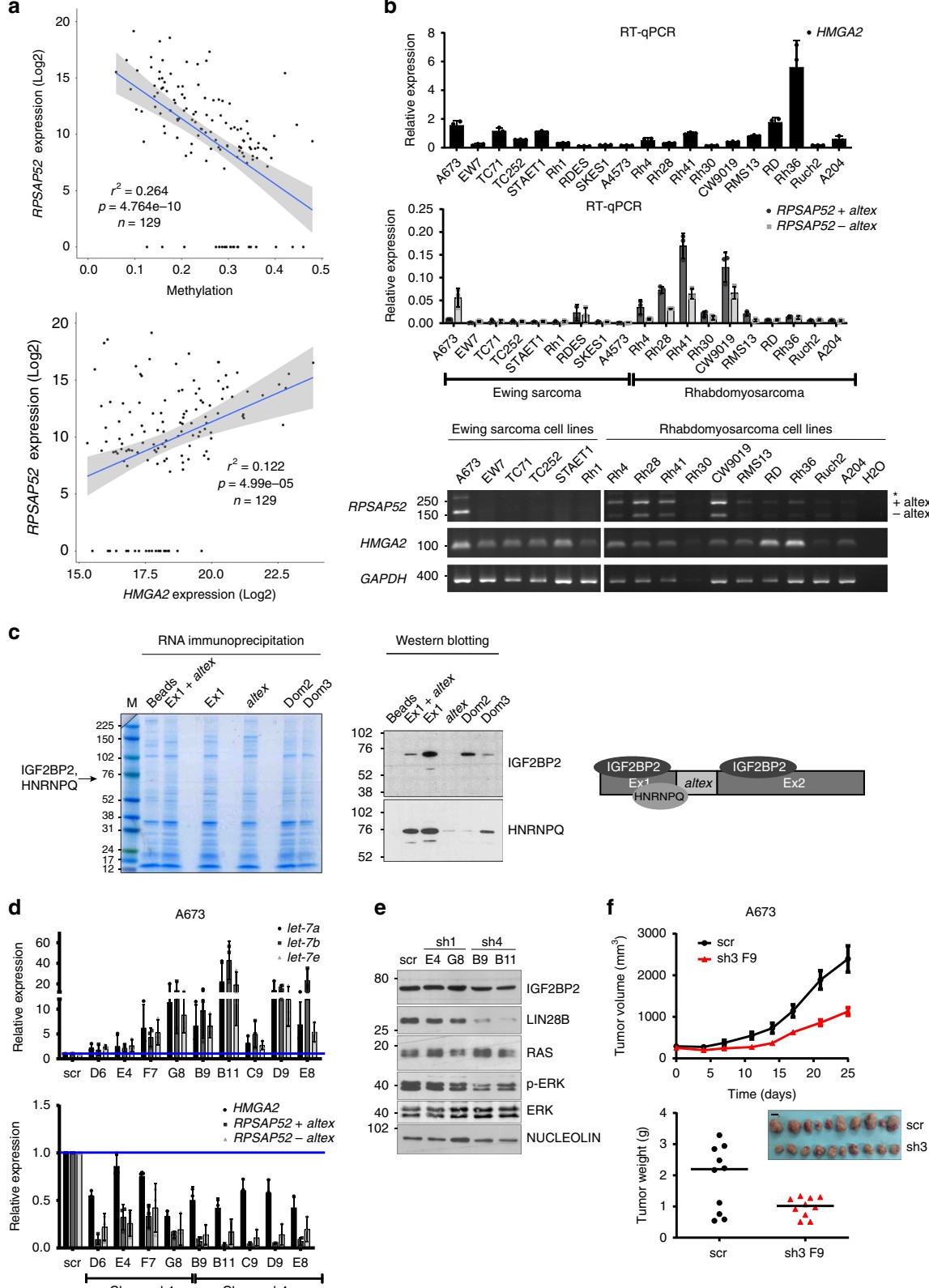

depletion of LIN28B does not impact on IGF2BP2 levels or its binding to mRNA targets, indicating that the regulation of *RPSAP52* on IGF2BP2 does not proceed through LIN28B (Supplementary Fig. 5h–j). Both *HMGA2* and *LIN28B* mRNAs are present at high levels upon *RPSAP52* depletion (Fig. 4d and Supplementary Fig. 4f), and their half-lives are not substantially

altered (Supplementary Fig. 5g), pointing to a decrease in their translation as a consequence of a diminished binding to IGF2BP2.

**RPSAP52 controls IGF2BP2 and mRNA distribution on polysomes.** To obtain direct evidence of the changes in translation efficiency for specific IGF2BP2 targets, we analyzed the

**Fig. 4** *RPSAP52* is abundantly expressed in sarcoma and regulates the LIN28B/*let-7* balance. **a** Upper graph: Pearson coefficient between *RPSAP52* expression levels and CGI methylation in the TCGA sarcoma cohort indicates a negative correlation. Lower graph: Pearson's index indicates a weaker association between *RPSAP52* and *HMGA2* expression levels in the same cohort. **b** Upper graphs: RT-qPCR analysis to estimate *HMGA2* and *RPSAP52* expression levels in a panel of Ewing's sarcoma and rhabdomyosarcoma cell lines. Expression is relative to *GUSB* mRNA levels. Graphs represent the mean ±SD of three independent RNA extractions. Lower panel: semi-quantitative RT-PCR analysis of expression in the same cell lines. The two *RPSAP52* isoforms are indicated, and the higher-migrating band depicted by an asterisk contains an additional exonic sequence (encompassing coordinates chr12:66,169,917–66,170,002 (hg19)), detected in those cells lines with the highest expression of *RPSAP52*. **c** RNA pull-down assays confirm the interaction of *RPSAP52* with IGF2BP2 and HNRNPQ in A673 cell extracts. Different truncated fragments of *RPSAP52* were assayed as indicated, and the band identified by MS and corresponding to IGF2BP2 and HNRNPQ is indicated on the protein gel (left). Western blot to test the association between *RPSAP52* RNA and IGF2BP2 and HNRNPQ proteins (middle panel). The drawing summarizes the data obtained from the pull-downs (right). **d** Total RNA from A673 stable clones constitutively expressing sh1 or sh4 shRNA sequences was analyzed by RT-qPCR to assess *HMGA2* and *RPSAP52* transcripts levels (lower graph) or *let-7* miRNAs levels (upper graph). Graphs represent the mean ±SD of three independent replicates. **e** Western blot on A673 clones to analyze protein levels upon stable knockdown of *RPSAP52* transcripts. **f** Growth-inhibitory effect of *RPSAP52* knockdown in A673 mice tumor xenografts. Upper graph: tumor volume ($n = 10$) was monitored over time. Mean values are shown ±SEM. Lower graph: tumors were excised and weighed at 25 days. The photograph shows the relative size of all tumors extracted. Scale bar = 10 mm. Source data are provided as a Source Data file

distribution of mRNAs across sucrose gradients in control or *RPSAP52*-depleted A673 cells. We observed no major changes in the polysome profiles of cells depleted of *RPSAP52* when compared with control cells, indicating that *RPSAP52* knockdown does not alter the global translational output of the cell (see gradient profiles in Fig. 6a–d and Supplementary Fig. 6a). Surprisingly, we detected a remarkable decrease in the amount of *HMGA2* and *LIN28B* mRNAs associated with polysomes (Fig. 6a, b), indicating a selective regulation in their translation as a function of *RPSAP52* expression. This was not observed for *NRAS* and *GAPDH* mRNAs (Fig. 6c, d). Analysis of total *HMGA2*, *LIN28B*, and *NRAS* mRNAs under the same conditions does not justify the specific redistribution of *HMGA2* and *LIN28B* across the gradients (Fig. 6e). These results indicate that the loss of binding to IGF2BP2 previously observed in iCLIP experiments correlates with lower translational efficiency for individual mRNAs, and we next asked whether IGF2BP2 protein itself is redistributed across the gradient. Importantly, even though IGF2BP2 coimmunoprecipitates with the same protein partners in pull-down experiments (Supplementary Fig. 6b, c), its co-sedimentation with translating poly-ribosomes is markedly reduced upon *RPSAP52* knockdown (Fig. 6f), demonstrating that *RPSAP52* expression mediates the recruitment of IGF2BP2 on polysomes. Taken together, the results suggest that the absence of the pseudogene decreases the recruitment of IGF2BP2 to large polysomes, thereby impacting on the translation of specific mRNAs.

**RPSAP52 alters key pathways and is a biomarker in sarcoma.** The influence of *RPSAP52* on IGF2BP2 binding affinity to its multiple mRNA targets might reflect the impact of this pseudogene on the control of several cellular processes. To identify such processes we interrogated general gene expression with an expression microarray platform under conditions of *RPSAP52* knockdown by shRNAs. As shown in Fig. 7a, 1% of the ~30,000 interrogated Entrez Gene RNAs were downregulated following knockdown, and 0.7% of the transcripts were upregulated. In agreement with a regulation mainly at the level of translation, none of the genes with differential IGF2BP2 binding along the 3′UTR, as seen by iCLIP-seq, is deregulated in the expression array analysis, indicating that (i) the differential binding observed is not a consequence of altered transcript expression, and (ii) none of the IGF2BP2 targets whose interaction with this protein is influenced by *RPSAP52* see their stability significantly altered as a consequence. However, they might participate in similar cellular pathways, since GO terms analysis indicated that the subset of downregulated genes was enriched in components of response to stimulus and signaling, whereas upregulated genes appeared more involved in

development (Fig. 7a). The full list of altered transcripts (fold change > 2, unpaired *t*-test *P*-value < 0.05) can be found in Supplementary Data 2. Among the downregulated genes, molecular functions that were overrepresented included genes involved in receptor binding and growth factor activity (e.g., *TIAM1*, *STYK1*, *AREG*, *MICB*) and sulfur compound binding (*CYR61* and *MGST1*). Of note, MGST1 is involved in the glutathione metabolism pathway and a marker of Ewing's sarcoma prognosis[40], high levels of NPY (a direct target of the EWS-FLI1 fusion) promotes the metastasis of Ewing's sarcoma models in vivo[41], CRABP1 and CPT1C favor tumor malignancy[42,43], and CD109 and PTPRZ1 are highly expressed in several cancer types (including sarcoma cell lines[44]) and promote stem cell-like properties[45]. Among the upregulated genes, genes involved in cytoskeletal protein binding were enriched and included MTSS1, a regulator of actin dynamics whose loss increases metastatic potential in a number of cancer types[46,47] (Supplementary Fig. 7a). The results from the expression arrays were validated by RT-qPCR under conditions of depletion of *RPSAP52* where *HMGA2* levels are largely unaffected (probably because the R-loop forming, nuclear *RPSAP52* is not effectively depleted by shRNAs), and with two different shRNAs that target distant regions on *RPSAP52* transcripts (Fig. 7b and Supplementary Fig. 7b). These results indicate that *RPSAP52* depletion implies a decrease in proliferative and self-renewal programs and suggests its potential as a biomarker in human samples. In support of this, patients with high *RPSAP52* expression levels had poorer prognosis than cases with low expression in the sarcoma patients cohort from TCGA database, whereas *HMGA2* expression did not show any prognostic effect in the same cohort (Fig. 7c). Since *RPSAP52* expression negatively correlates with hypermethylation of the associated CpG island (Fig. 4a), methylation itself is also a marker of better prognosis (Fig. 7d), reinforcing the relevance of considering lncRNA expression regulation in translational medicine.

Taken together, our results suggest that the pseudogene *RPSAP52* controls the HMGA2/IGF2BP2/LIN28B axis through a double mechanism that involves, in the nucleus, the positive transcriptional regulation of *HMGA2*, and in the cytoplasm, the regulation of the function of IGF2BP2 protein as a translational co-regulator (among others, of *LIN28B* and *HMGA2* mRNAs), which in turn results in a downregulation of *let-7* miRNAs and derepression of their pro-proliferative targets (see diagram depicting our working model in Fig. 7e). *RPSAP52* thus displays characteristics of an oncogenic gene whose dysregulation might contribute to the progression of a number of human cancers.

## Discussion
The detailed mechanism by which lncRNAs may contribute to altering the output of signal transduction pathways is largely

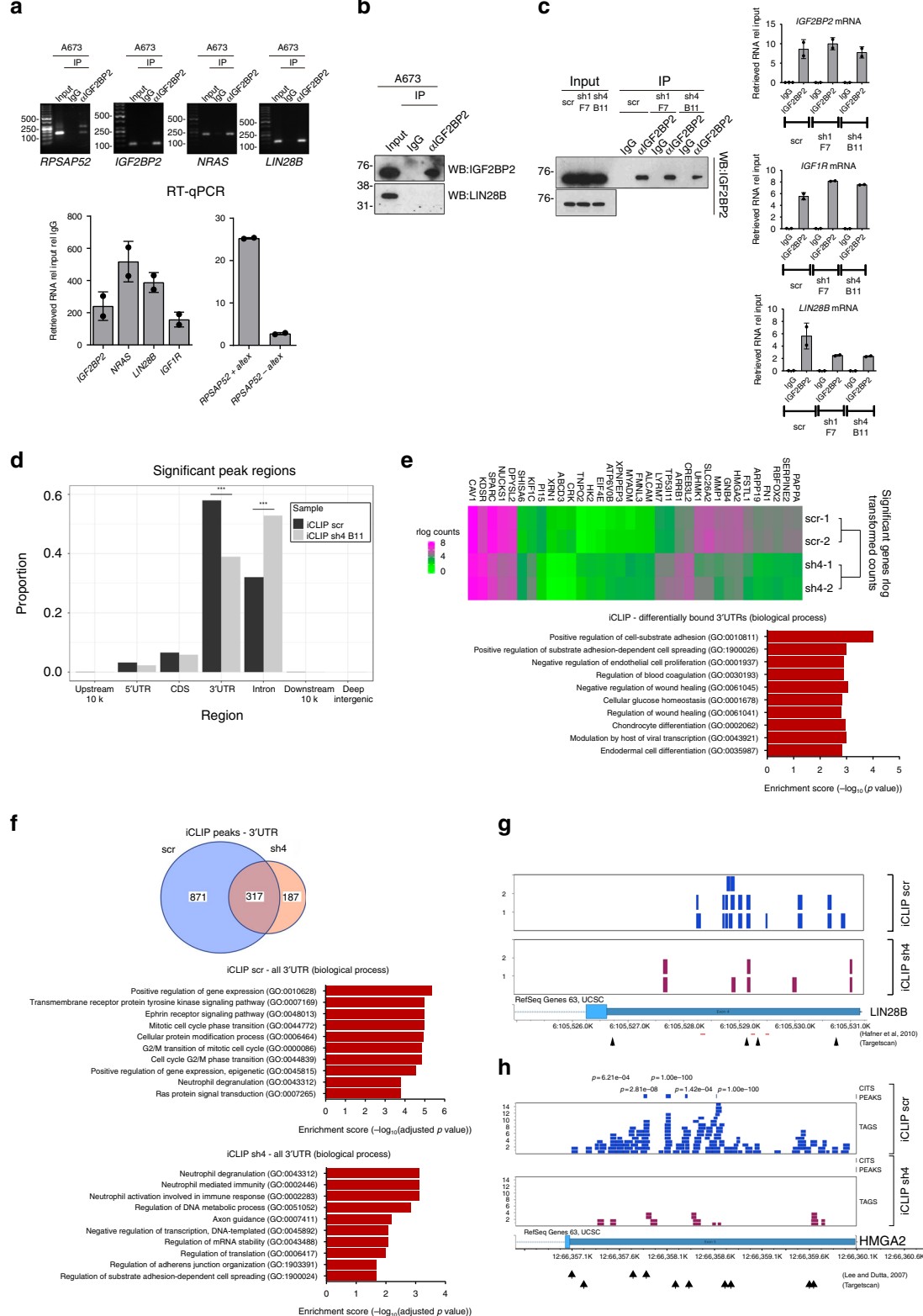

unexplored. Our previous work had shown that the transcribed pseudogene *RPSAP52* enhances *HMGA2* transcription through the formation of an R loop structure[23]. We have further explored the impact of *RPSAP52* expression in cell physiology and propose a mechanism of action that also influences post-transcriptional regulation in the cytoplasm through the interaction with the RNA binding protein IGF2BP2. Regulation of IGF2BP2 expression or

function by lncRNAs appears as a common theme in a number of lineage commitment programs, including adipocyte, cardiac or muscle differentiation[48–50]. However, while other studies have reported lncRNAs that interact with IGF2BP2 and compete for its binding to target mRNAs (e.g., *LncMyoD* promotes muscle differentiation by outcompeting *c-Myc* and *N-Ras* mRNAs for IGF2BP2 binding[48]), in the cancer setting that we are studying

**Fig. 5** Binding of IGF2BP2 to its mRNA targets is affected by *RPSAP52* knockdown. **a** IGF2BP2 was immunoprecipitated from A673 extracts and the pulled-down RNA was analyzed by RT-PCR. The *IGF2BP2* and *NRAS* mRNAs were used as positive controls. Gel images represent semi-quantitative RT-PCR, whereas data on graphs represent means of two independent RT-qPCR analysis ±SD. **b** Immunoprecipitation of IGF2BP2 from A673 extracts followed by western blot of retrieved proteins. 10% of total extract prior to IP was loaded as control (input). **c** IGF2BP2 was immunoprecipitated in control or *RPSAP52*-depleted A673 cells, and the retrieved proteins and RNAs were isolated and analyzed by western blot (left) or RT-qPCR (right), respectively. Graphs correspond to means from two replicates ±SD. **d** Analysis of IGF2BP2 binding targets from iCLIP-seq experiments in control (scr) or depleted cells (sh4 B11). The absolute number of peaks mapping to 3′UTR regions were 1762 (scr) and 622 (sh4), and to intronic regions were 973 (scr) and 846 (sh4). Asterisks correspond to *P*-values < 2.2e−16 (two-tailed Fisher's tests). **e** Above: heatmap of genes with differential iCLIP counts on their 3′UTR. Results from two experiments are shown. Below: differential enrichment of these genes according to GO biological process categories (top ten are shown). **f** Above: Venn diagram showing the relation between genes with significant iCLIP peaks present on their 3′UTR regions in the control (scr) or *RPSAP52*-depleted (sh4) sample. Below: top ten GO enrichment categories (Biological process) for the same genes in the scr or sh4 sample. **g** UCSC Genome Browser view of *LIN28B* 3′UTR with the read coverage from IGF2BP2 iCLIP experiment. Previous IGF2BP2-CLIP data positions are shown in red, and predicted *let-7* binding sites are indicated by the arrows. **h** UCSC Genome Browser view of *HMGA2* 3′UTR with the read coverage from IGF2BP2 iCLIP experiment. Position of significant peaks and CITS are shown above the profiles, and predicted *let-7* binding sites are indicated by the arrows. Source data are provided as a Source Data file

reexpression of *RPSAP52* facilitates IGF2BP2 binding to a subset of mRNA targets, prominently *HMGA2* and *LIN28B* mRNAs. Our data indicate that this is achieved through modulation of the binding affinity that IGF2BP2 has for particular 3′UTRs and its distribution in large polysomes. This is reminiscent of the mechanism of action of *HIF1A-AS2* in glioblastoma cell lines, where binding of an antisense transcript to IGF2BP2 and DHX9 stimulates expression of their target mRNAs and promotes adaption to hypoxic stress[51]. Thus, our working model is that by forming ternary complexes (IGF2BP2-*RPSAP52*-other mRNAs), *RPSAP52* may influence the recruitment into ribonucleoprotein particles that dictate mRNA fate, and in particular enhance the translation of mRNAs that would otherwise be repressed by miRNAs. Binding by IGF2BP3 (another member of the IGF2BP family), for instance, has been associated with resistance to miRNA-dependent destabilization for many oncogenes, including *HMGA2* and *LIN28B*[52]. We hereby describe a similar scenario for IGF2BP2, and IGF2BPs are thus emerging as key nodes that integrate lncRNA-mediated post-transcriptional regulation of gene expression and pro-proliferative and self-renewal axis.

An important added layer of regulation exerted by *RPSAP52* is the influence on *let-7* levels, which may be a consequence of the control on LIN28B translation efficiency, or (in those cells where LIN28B is absent, such as MCF10A), may derive from the altered levels observed in IGF2BP2 protein itself upon depletion of the pseudogene. In fact, in glioblastoma cells lacking LIN28, *let-7* targets have been observed to be protected from miRNA-dependent silencing by the binding of IGF2BP2 to *let-7* miRNA responsive elements[27]. This in turn may indirectly cause a decrease of *let-7* levels, since miRNA turnover might also depend on binding to mRNA targets, with some previous evidence suggesting that target availability prevents miRNA decay[53,54]. The greater effect of directly inhibiting biogenesis versus indirectly influencing the turnover might explain why *let-7* levels increase moderately in MCF10A upon *RPSAP52* depletion (where LIN28B is not expressed and LIN28A is not altered, but IGF2BP2 levels decrease), and, by contrast, increase by almost one order of magnitude more in A673 cells (where expression of LIN28B protein is reduced) (compare Figs. 2c and 4d). Taken as a whole, *RPSAP52* is a pseudogene with an important impact on a major tumor suppressor miRNA. While silencing of *HMGA2* expression by *let-7* has been reported before[55], this is the first time that regulation of *let-7* levels by transcripts originating from *HMGA2 locus* is proposed. Importantly, this effect does not proceed through *HMGA2* itself, since depletion of *HMGA2* expression with gapmers actually increases *RPSAP52* levels and consequently results in a decrease in *let-7* family (Fig. 2e). This

adds further complexity to the regulatory network, one hypothesis being that the tumorigenic cell activates an alternative pathway (increase of *RPSAP52*) to compensate for the loss of HMGA2 function.

Consistent with their convergent roles in the same pathway, low expression of *HMGA2/RPSAP52* in differentiated cells and reexpression in cancer mirrors LIN28 levels, which is one of the key players in maintenance of the pluripotent state. *Let-7* levels are maintained low in embryonic stem cells and certain primary tumors due to inhibition by LIN28 proteins, which are present at characteristically high levels in undifferentiated cells[56,57]. Of all tumor suppressor miRNAs, *let-7* is the one whose loss is most frequently correlated with poor prognosis in meta-analysis reports[58]. Accordingly, LIN28A/B high expression is a marker of poor prognosis and more aggressive tumors in a variety of cancers, and their levels have also been associated with metastatic and drug-resistant cases[59]. Thus, regulation of LIN28B/*let-7* balance is one important driver in cancer development.

An important aspect of this LIN28B/*let-7* balance is their counteracting action on the stemness characteristics of cancer cells. Interestingly, LIN28B/*let-7* signaling has been shown to regulate endogenous Oct4 and Sox2 expression by using ARID3B and HMGA2 as downstream effectors, and thereby regulate stemness properties in oral squamous cancer[60]. Also, the role of *let-7* in antagonizing self-renewal and promoting differentiation has been established via targeting of Myc, Ras, and HMGA2 pathways[61,62]. In accordance with *let-7* anti-pluripotency properties, we observe a decrease in NANOG and OCT4 levels as well as in clonogenicity upon *RPSAP52* depletion (Fig. 3d, e), suggesting that *RPSAP52* is an enhancer of stem cell characteristics. To date, few lncRNAs have been thoroughly described regarding their involvement in stemness, among them *H19* (whose down-regulation reduces NANOG, OCT4, and SOX2 in glioma and breast cancer[63]) and lncRNA *ROR* (which inhibits proliferation of glioma stem cells by negatively regulating KLF4[64]). In particular, *H19* has recently been proposed to facilitate tumorigenesis through sponging of *let-7*[65]. The regulatory mechanism used by *RPSAP52*, by contrast, targets *let-7* family through modulation of LIN28B and/or target availability.

Taken together, we have observed that *RPSAP52* (1) stimulates proliferative and self-renewal axes together with a reduction of *let-7* levels, (2) promotes tumorigenic behavior in vitro and in vivo, and (3) is overexpressed in a number of human cancers and its expression is associated with worse outcome. This, together with its virtual absence in normal differentiated cells and embryonic expression pattern allows us to propose that *RPSAP52* is an oncofetal pseudogene that enhances proliferative

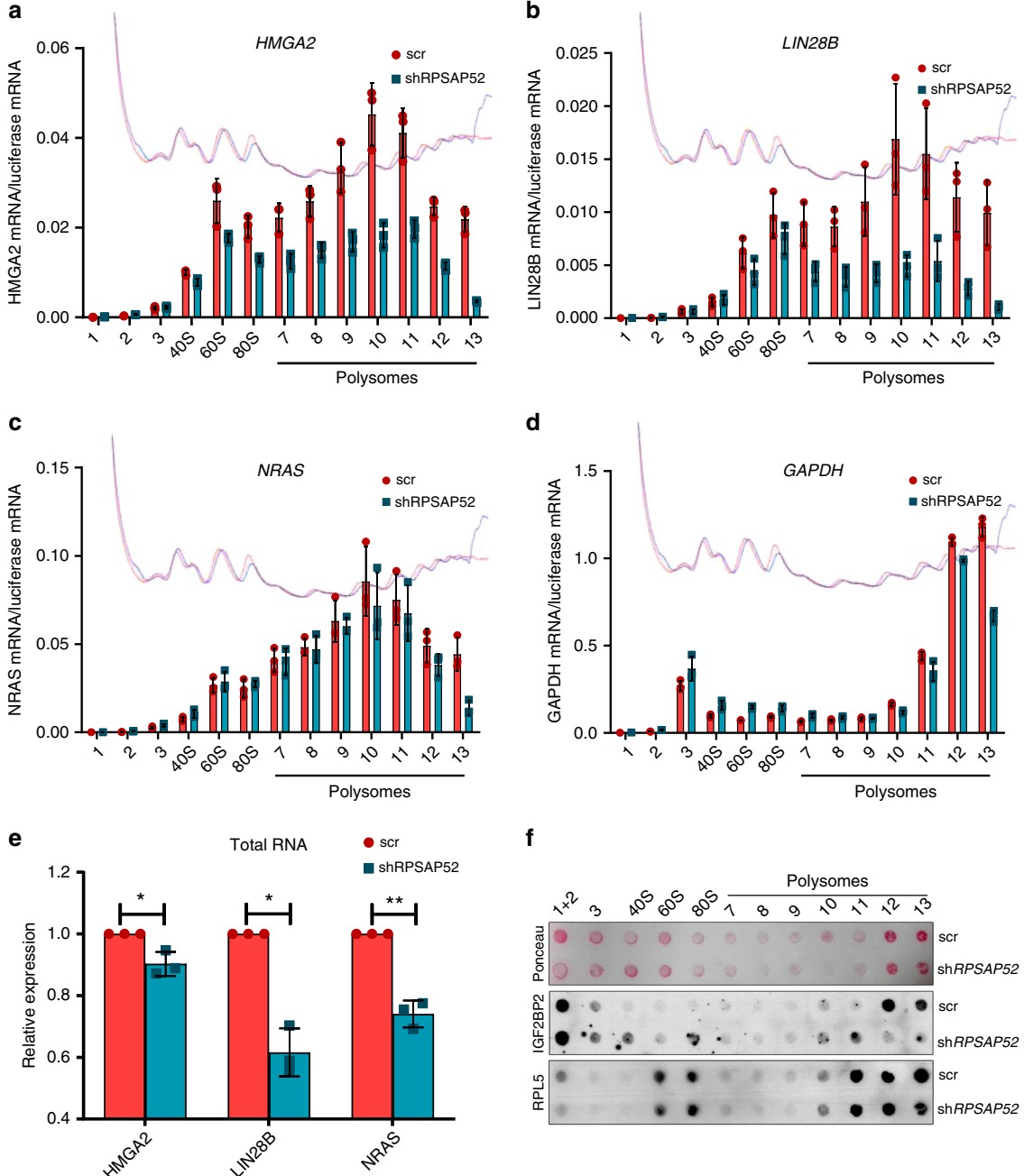

**Fig. 6** IGF2BP2 is redistributed on polysome gradients upon *RPSAP52* depletion. **a–d** Polysome profiles of A673 cells stably depleted of *RPSAP52* (shRPSAP52, corresponding to sh4 B11 clone) or control cells (scr). *HMGA2* mRNA (**a**), *LIN28B* mRNA (**b**), or *NRAS* mRNA (**c**) distribution across the gradient was evaluated in each fraction by RT-qPCR. For comparison, *GAPDH* mRNA distribution was also assessed (**d**). Graphs represent the mean ±SD of three replicates. The red and blue lines indicate absorbance at 260 nm for each fraction in control or depleted cells, respectively. **e** Total RNA from the same cells (*n* = 3) was analyzed by RT-qPCR. Mean values are shown ±SD. A two-tailed student *t*-test was used (*$P < 0.05$, **$P < 0.01$). **f** Protein extracted from the 20% of the polysome profile fractions shown in (**a-d**) were subjected to dot blot analysis with an anti-IGF2BP2 antibody (middle panel) or with anti-RPL5 antibody as control (lower panel). Proteins from 10% of fractions 1 and 2 were loaded together. Membranes were previously stained with Ponceau S (top panel) for loading control. Source data are provided as a Source Data file

and survival programs across several tumor types and whose expression in cancer can have important clinical implications. Indeed, *RPSAP52* levels are more useful as biomarkers in sarcoma than *HMGA2* mRNA levels, which do not seem to correlate well with protein levels (as suggested by our work and others[66]), probably due to the complex post-transcriptional regulation of HMGA2. The potential use of this pseudogene as an effective therapeutic target in human cancer will thus be the focus of future studies.

## Methods

**DNA methylation analysis**. Genome-wide DNA methylation analysis was performed with the 450K DNA methylation microarray from Illumina (Infinium HumanMethylation450 BeadChip). Bisulfite-treated DNA from the indicated breast cancer cell lines was hybridized onto the array. A three-step normalization procedure was performed using the lumi package v2.30.0 (available for Bioconductor, within the R v3.4.3 statistical environment), consisting of color bias and background level adjustment and quantile normalization across arrays. The methylation level ($\beta$-value) of CpG sites was calculated as the ratio of methylated signal divided by the sum of methylated and unmethylated signals.

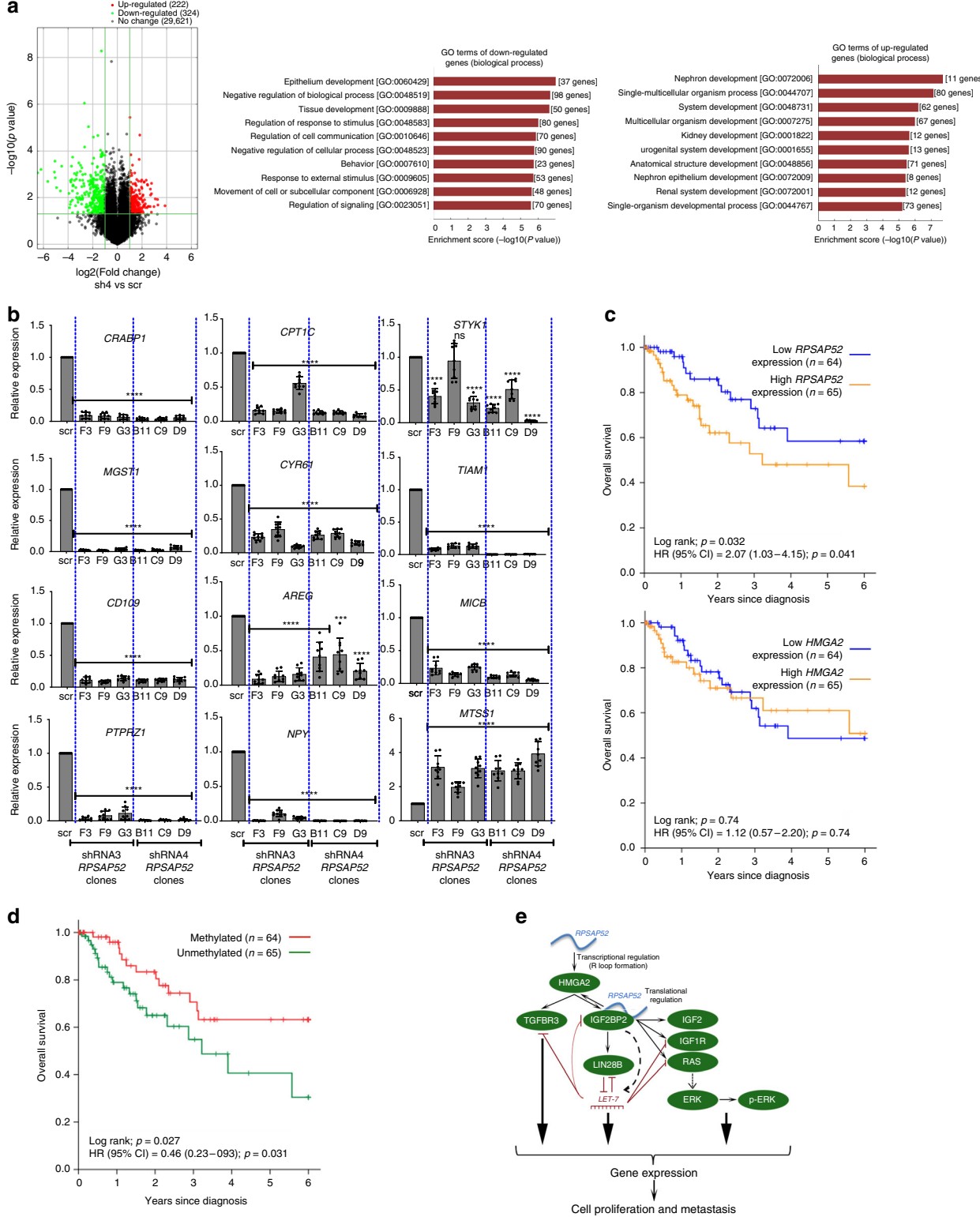

**Bisulfite genomic sequencing**. The Methyl Primer Express v1.0 software (Applied Biosystems) was used to design specific primers for the methylation analysis of *HMGA2/RPSAP52* island (Supplementary Table 1). Genomic DNA (1 µg) was subjected to sodium bisulfite treatment using the EZ DNA Methylation-Gold kit (Zymo Research). For bisulfite genomic sequencing, 300–500 bp fragments were amplified using 1–2 µl of bisulfite-converted DNA with Immolase Taq polymerase (Bioline) for 42 cycles. The resulting PCR products were gel-purified with NucleoSpin® Gel and PCR Clean-up (Macherey-Nagel) and then cloned into the pSpark® TA vector (Canvas) according to the manufacturer's protocol. For all samples, 10 colonies were randomly chosen, the DNA was purified using NucleoSpin® 96 Plasmid (Macherey-Nagel) and sequenced by the 3730 DNA Analyzer (Applied Biosystems). After sequencing analysis with BioEdit v7.2.5 software, C nucleotides that remained unaltered were transformed into percentages of CpGs showing methylation.

**Western blotting**. Cell pellets were resuspended in lysis buffer (50 mM Tris-HCl pH 8, 5 mM EDTA, 350 mM NaCl, 0.5% NP40, 10% glycerol, 0.1% SDS and phosphatase inhibitors), sonicated and centrifuged to recover the supernatant. The concentration

**Fig. 7** *RPSAP52* expression influences proliferative cellular programs and is a prognosis factor. **a** Left: Volcano plot indicating differential expression (green = down, red = up) between control (scr) and *RPSAP52*-depleted (sh4) A673 cells. The vertical green lines correspond to 2.0-fold up and down, respectively, and the horizontal green line represents a *P*-value of 0.05 (two-tailed unpaired *t*-test). Right: enriched GO terms for shRNA-*RPSAP52*-affected genes. The *y* axis shows GO terms and the *x* axis shows statistical significance (two-tailed Fisher's exact test). **b** RT-qPCR analysis of candidate genes altered in *RPSAP52*-depleted A673 cells. Clones stably expressing two different shRNAs were analyzed. Graphs represent the mean ±SD of three replicates (two-tailed unpaired student *t*-test, \*\*\**P* < 0.001, \*\*\*\**P* < 0.0001). **c** Above: in the TCGA sarcoma cohort, Kaplan–Meier analysis of overall survival indicates that patients with high *RPSAP52* expression levels have poorer prognosis than cases with low expression. Below: *HMGA2* expression has no prognostic value in the same cohort. Significance of the log-rank test is shown. **d** Kaplan–Meier analysis of overall survival in the sarcoma cohort from TCGA, indicating that patients with a hypermethylated *HMGA2/RPSAP52* promoter display better prognosis. **e** Summary of the results in the context of HMGA2/IGF2BP2/*let-7* axis. *RPSAP52* positively regulates *HMGA2* expression through both transcriptional and post-transcriptional mechanisms. Binding of *RPSAP52* to IGF2BP2 in the cytoplasm might promote downregulation of *let-7* levels by LIN28B-dependent and independent mechanisms. This binding could also modulate the formation of mRNPs for a number of IGF2BP2 mRNA targets, thereby directing their translation efficiency. Source data are provided as a Source Data file

was determined with the Pierce BCA Protein Assay Kit (#23227, ThermoFisher). Proteins were boiled for 5 min with Laemmli buffer (2% SDS, 10% glycerol, 60 mM Tris-Cl pH 6.8, 0.01% bromophenol blue) plus 2% 2-mercaptoethanol as a loading buffer, and equal amounts of extracts were loaded onto Tris-Glycine-SDS gels. Proteins were transferred to a nitrocellulose membrane (Whatman, GE Healthcare) and incubated overnight at 4 °C with primary antibodies diluted in 5% skimmed milk in PBS containing 0.1% Tween-20. The detected proteins were IGF2BP2 (H00010644-M01, Abnova, 1:500), RAS (ab55391, Abcam, 1:500, which recognizes all RAS proteins), LIN28B (ab71415, Abcam 1:1000), IGF1R (#3027, Cell Signaling, 1:1000), ERK (#4695, Cell Signaling, 1:1000), p-ERK (#9101, Cell Signaling, 1:1000), LAMIN B1 (ab16048, Abcam, 1:4000), LIN28A (#8641, Cell Signaling, 1:750), α-TUBULIN HRP (ab40742, Abcam, 1:5000), HNRNPQ (ab184946, Abcam, 1:10,000), β-ACTIN HRP (a3854, Sigma, 1:20,000), NANOG (#4903, Cell Signaling, 1:2000), OCT4 (#2750, Cell Signaling, 1:1000), SOX2 (#4195, Cell Signaling, 1:1000), NUCLEOLIN (#8031, Santa Cruz, 1:2000), HMGA2 (ab97276, Abcam, 1:1000), HISTONE H3 (ab1791, Abcam, 1:5000), RPSA (ab133645, Abcam, 1:1000), and RPL5 (A303-933A, Company Bethyl, 1:1000). After three washes with PBS containing 0.1% Tween-20, membranes were incubated for 1 h at room temperature in a bench-top shaker with the secondary antibodies conjugated to horseradish peroxidase anti-rabbit IgG (A0545, Sigma, 1:10,000) or anti-mouse IgG (NA9310, GE HealthCare, 1:5000). ECL reagents (Luminata-HRP; Merck-Milllipore) were used to visualize the proteins.

**Nuclear/cytoplasmic fractionation and poly(A) selection.** Subcellular fractionation was performed with PARIS™ kit (#AM1921, Life Technologies). Equal amounts of RNA from each fraction were subject to RT-qPCR and the results were normalized taking into account the total quantity of RNA recovered from each fraction. To verify the nuclear and cytoplasmic fractionation of the mRNA, *RNU6B* and *GAPDH* were used as controls. The separation was confirmed at the protein level by western blot with HISTONE H3 (ab1791, Abcam, 1:5000) and α-TUBULIN HRP (ab40742, Abcam, 1:5000). Poly(A) + and poly(A)− RNAs were separated using the Dynabeads® mRNA Purification kit (#61006, Life Technologies), using three rounds of selection. RNA enrichment in each fraction was then analyzed by RT-qPCR, using *GAPDH* and *RNU6B* as controls.

**RNA-biotin pull-down.** Full-length *RPSAP52* (including alternative exon) or truncated fragments, as well as the antisense version or the sequence corresponding to the unrelated Uc.160 + RNA, were biotin-labeled by standard in vitro transcription reactions and gel-purified. DNA templates for transcription were prepared by PCR with oligos described in Supplementary Table 1. The pull-downs were carried out with 10 pmol of each biotinylated RNA and 1 mg of total MCF10A or A673 protein extracts. Following incubation with the extract, each RNA was retrieved with 25 μl of Dynabeads® M-270 Streptavidin beads (#65305, Invitrogen) and washed in RIP buffer (150 mM KCl, 25 mM Hepes at pH 7.9, 5 mM EDTA, 0.5 mM DTT, 0.5% NP40, 1× protease inhibitor cocktail (Roche)). Binding proteins were released through boiling in SDS loading buffer and samples were run on a 4–12% gradient pre-cast Bis-Tris protein gels (Invitrogen) in MOPS buffer. After electrophoresis, the gels were either stained with SYPRO Ruby (Invitrogen) for band visualization and MS analysis or transferred to nitrocellulose membranes for western blotting.

**In-gel digestion and LC-MS/MS analysis.** Gel bands were manually excised and digested with trypsin overnight. Bands were then washed with water, 50 mM ammonium bicarbonate and 50% acetonitrile. Samples were subsequently reduced with 10 mM DTT and alkylated with 35 mM iodoacetamide. Extracted peptides were analyzed on a Ion Trap Amazon Speed ETD (Bruker Daltonics, Bremen, Germany) fitted with a captivespray source (Bruker, Daltonisc) following separation with Easy-nLCII apparatus (Proxeon). Peptides were separated in a reverse phase chromatography using a nano-capillary analytical c18 column. Peptide masses were analyzed at full scan MS, and then at MS/MS fragmentation for the

most intense peaks. Data were analyzed using the Mascot search engine and the SwissProt human database.

**Reverse pull-down.** Endogenous IGF2BP2 was immunoprecipitated from A673 cell extracts. One milligram of total protein was incubated overnight with 2 μg of anti-IGF2BP2 polyclonal antibody (#H00010644-M01, Abnova) or control mouse IgG antibody (#12-371, Millipore) and 40 μl of Dynabeads® M-280 anti-mouse IgG beads (#11202D, ThermoFisher) in 1 ml of RIP buffer (150 mM KCl, 25 mM Hepes at pH 7.9, 5 mM EDTA, 0.5 mM DTT, 0.5% NP40, 1× protease inhibitor cocktail (Roche)). Beads were then washed three times with RIP buffer and 10% of the volume was boiled in the presence of Laemmli buffer for western blot analysis. The pulled-down RNA in the remaining 90% of beads was extracted by adding 1 ml of TRIzol® Reagent (15596-018, ThermoFisher). After phenol extraction and isopropanol precipitation, the final pellet was resuspended in 10 μl of $H_2O$ and ret-rotranscribed. cDNA was analyzed by either 30 cycles of semi-quantitative RT-PCR or by RT-qPCR (Applied Biosystems 7900HT Fast Real-Time PCR System). Two micrograms of input RNA was processed in parallel to estimate pull-down efficiency.

**Cell culture.** MCF7, MCF10A, HCC1143, and A673 cell lines were purchased from ATCC. The remaining breast and sarcoma cell lines were obtained from Dr. Esteller and Dr. Tirado's labs, respectively. Authenticity of the cell lines was routinely confirmed by STR profiling analysis done at qGenomics SL (Esplugues de Llobregat, Barcelona, Spain). All cell lines were routinely checked for mycoplasma contamination. Non-malignant MCF10A breast cells were grown in DMEM/Ham's F-12 medium (#L0093-500, Biowest) supplemented with 20 ng ml$^{-1}$ EGF (#SRP3027, Sigma), 500 ng ml$^{-1}$ hydrocortisone (#H0888, Sigma), and 10 μg ml$^{-1}$ insulin (#I9278, Sigma). Ewing's sarcoma A673 cells and breast cancer HCC1143 cell line were grown in RPMI-1640 medium with GlutaMAX (#61870-010, Gibco). HEK293T cells, used for the production of lentiviral particles, and breast cancer Hs578T cell line were cultivated in DMEM with GlutaMAX (#31966-021, Gibco). All the media were supplemented with 10% fetal bovine serum (FBS) (#10270, Gibco), and the cells were grown at 37 °C in a humidified atmosphere of 5% $CO_2$ and 95% air.

**Plasmid construction and transfections.** Stable knockdown of *RPSAP52* was achieved with the following sequences: shRNA1 and shRNA4 target, respectively, the 5′TCCTTAAGCTCCTTGCAGT3′ and 5′CACGGACTCTTAAGCAACA3′ sequences of *RPSAP52* mRNA (both located on the last exon), whereas shRNA3 targets the 5′GTGCAAGACTCAGGAGCTA3′ sequence of *RPSAP52* (on the first intron, which is inefficiently spliced). These shRNAs were expressed by cloning oligos shRPSAP52-1for and shRPSAP52-1rev (shRNA1), shRPSAP52-4for and shRPSAP52-4rev (shRNA4), and shRPSAP52-3for and shRPSAP52-3rev (shRNA3) into the BamHI and EcoRI sites of the vector pLVX-shRNA2 (Clontech). A scramble (scr) sequence was used as a control. For lentivirus-mediated depletion, HEK293T cells were transfected with pLVX-shRNA2-constructs plus packaging plasmids with jetPRIME® (Polyplus-transfection) according to the manufacturer's recommendations. The target cell line was infected with the supernatant containing viral particles 48 h post-transfection. ZsGreen1 was used as a marker to visualize transductants by fluorescence microscopy, and these cells were selected by fluorescence-activated cell sorting (FACS) and plated to obtain stable clones. Antisense oligonucleotides (LNA™ GapmeRs, #300600, Exiqon) targeting *HMGA2* mRNA (HMGA2), exon1 (RPSAP52 Ex) or the first intron (RPSAP52 RL1 and RL2) of *RPSAP52* transcript were transfected to a final concentration of 65 nM using HiPerfect (Qiagen). Cells were retransfected 48 h later and collected 72 h after the second round of LNA treatment. A control LNA GapmeR (#300610, Exiqon) was used as mock transfection. For siRNA-mediated knockdown of LIN28B, cells were transfected with a 1:1 mix of two different siRNAs against LIN28B (#216387-216388, Ambion) and a negative control (C-) (#AM4611, Ambion), using Lipofectamine™ RNAiMAX Transfection Reagent

(#13778, Invitrogen) according to the manufacturer's recommendations. The overexpression of LIN28B was achieved with pcDNA3-FLAG-Lin28B (#51373, Addgene), and we used pcDNA3.1 + (Invitrogen) as a control (empty) and jet-PRIME® (Polyplus-transfection) as the transfection reagent.

**Transcription/translation assay.** *RPSAP52* full-length transcript (from RefSeq NR_026825 annotation) was amplified by PCR from A673 cDNA with oligos T7-RPSAP52for-TnT and RPSAP52rev-TnT, producing two isoforms that include or exclude the alternative exon downstream of the T7 bacteriophage promoter. After gel purification, the PCR product was used directly in coupled transcription and translation reactions in reticulocyte extracts (TNT® Quick Coupled Transcription/Translation Systems, Promega), following the manufacturer's indications and by labeling the reactions with $^{35}$S. The translation products were separated by SDS-PAGE, the gel was then vacuum-dried and exposed overnight with an autoradiography film.

**RNA isolation and RT-qPCR analysis.** Total RNA, including miRNAs, was extracted using the Maxwell® RSC instrument with the Maxwell® RSC miRNA Tissue kit (Promega) according to the manufacturer's recommendations. For mRNA expression analysis, total RNA was reverse transcribed using the Super-Script$^{TM}$ III Reverse Transcriptase (#18080, Invitrogen). Real-time PCR reactions were performed in triplicate in an Applied Biosystems 7900HT Fast Real-Time PCR system, using 30–100 ng cDNA, 6 μl SYBR® Green PCR Master Mix (Applied Biosystems), and 416 nM primers in a final volume of 12 μl for 384-well plates. All data were acquired and analyzed with the QuantStudio Design & Analysis Software v1.3.1 and normalized with respect to *GUSB* as endogenous control. Relative RNA levels were calculated using the comparative Ct method (ΔΔCt). For miRNA expression analysis, miRCURY LNA™ Universal RT microRNA PCR System (Exiqon) was used, according to the manufacturer's recommendations, with the Universal cDNA Synthesis Kit II (#203301) for the RNA retrotranscription and the ExiLENT SYBR® Green master mix (#203421) for the RT-qPCR in the Light-Cycler® 480 (Roche) with the LightCycler® 480 Software v1.5.0 SP4. To normalize the data, *RNU6B* or *miR-195* were used as endogenous control.

**Actinomycin D treatment and RNA stability analysis.** Control or *RPSAP52*-depleted A673 clones were treated with either 0.5% DMSO or 5 μg ml$^{-1}$ Actino-mycin D (Sigma) for 9 h. Pellets of each condition and treatment were harvested at different times and RNA was extracted for the RT-qPCR experiments. All data were normalized with respect to *GUSB* as endogenous control and gene expression fold-changes induced by Actinomycin D were calculated relative to the control (DMSO) cells of each condition and time point. *c-FOS* and *GAPDH* were used as controls of the experiment due to their short and long half-life, respectively.

**SRB assay.** Cell viability and proliferation were determined by the sulforhodamine B (SRB) assay. Cells were seeded in 96-well microplates in medium with 10% FBS, and the experiment started after 24 h of incubation at 37 °C and 5% CO$_2$. The optimal cell number (100 cells per well for MCF10A and 2000 cells per well for Hs578T and HCC1143) was determined to ensure that the cells were in growth phase at the end of the assay. During 7 consecutive days, at least 6 wells per condition were processed as follows: the medium was removed and the cells were fixed with 10% trichloroacetic acid for 1 h at 4 °C. Then, two washes with 1% acetic acid were performed and the viable cells were stained with 0.057% SRB in 1% acetic acid. Following 30 min of incubation at RT, the SRB was removed by washing twice with 1% acetic acid. The wells were air-dried completely and the SRB bound to the viable cells was dissolved with 100 μl of Tris-HCl 10 mM (pH 10.0). Absorbance at 540 nm was determined on an automated microtiter plate reader PowerWave XS (BioTek).

**Colony formation assay.** Cells were seeded into 35 mm dishes with three triplicates per condition at a density of 200 cells per plate for MCF10A and HCC1143 and 500 cells per plate for Hs578T. They were maintained for 8–15 days in a humidified incubator with 5% CO$_2$ at 37 °C. Cells were then fixed in 4% paraformaldehyde and stained with 0.5% crystal violet for 30 min. Digital images were obtained using GBox (Syngene) and colonies containing more than 50 cells were counted manually using ImageJ v1.50 software. Plating efficiency and survival fractions were determined by using the following formulas:

$$\text{Plating efficiency} = \frac{\text{number of colonies obtained}}{\text{number of cells seeded}}$$

$$\text{Surviving fraction} = \frac{\text{plating efficiency}}{\text{number of colonies obtained in the control condition}} \times 100$$

**Real-time migration assay.** The xCELLigence Real-Time system (ACEA Biosciences) was used with CIM-16 plates of 8 μm pore membranes. The lower chamber wells were filled with 160 μl of medium containing 10% FBS and the top chamber wells with 40 μl of serum-free medium. The two chambers were assembled together and allowed to equilibrate for 1 h at 37 °C and 5% CO$_2$. Cells were incubated for 24 h in serum-free medium, rinsed with PBS, trypsinized and resuspended in medium supplemented with 10% FBS to inactivate the trypsin,

followed by centrifugation and resuspension in serum-free medium. A total of $8 \times 10^4$ cells were seeded onto the top chamber of CIM-16 plates and placed into the xCELLigence system for data collection after background measurement. The software RTCA 2.0 was set to collect impedance data every 15 min. The cell index represents the capacity for cell migration, whereas the slope of the curve can be related to the cell invasion ability.

**Transwell migration assay.** Transwell® Permeable Supports (#3422, Cultek) with 8 μm pore polycarbonate membranes in 24-well plates were used to measure cell migration. Cells were incubated for 24 h in serum-free medium, rinsed with PBS, trypsinized, and resuspended in 10% FBS-containing medium to inactivate the trypsin, followed by centrifugation and resuspension in serum-free medium. A total of $1 \times 10^5$ cells were seeded onto each transwell with 150 μl of serum-free medium and the transwells were placed in the wells of a 24-well plate with 500 μl of 10% FBS-containing medium. The chemoattractant promoted the migration of the cells from the upper part of the transwell to the lower part. After 24 h of incubation at 37 °C and 5% CO$_2$, the cells in the upper part of the membrane were removed with a cotton swab and several washes with 1× PBS. Cells in the lower part were fixed for 10 min with ice-cold 100% methanol. For the staining, cells were covered with 0.5% crystal violet in 25% methanol for 10 min. Transwells were washed several times with 1× PBS and air-dried. Membranes were then mounted on a slide for image acquisition.

**Clonogenicity assay.** The clonogenicity of *RPSAP52*-depleted MCF10A clones was tested in soft agar by using the CytoSelect 96-well Cell Transformation Assay Kit (Cell Biolabs, #CBA-130), following the manufacturer's instructions. Briefly, a base agar layer was prepared by mixing equal volumes of 1.2% agar solution and 2× DMEM/20%FBS medium in each well of a 96-well flat-bottom microplate. In total, 5000 cells per well were seeded in a top layer by mixing equal volumes of the cell suspension, 1.2% agar solution and 2× DMEM/20%FBS (1:1:1), and incubated for 6 days after covering the solidified cell agar layer with 100 μl of DMEM-F12 medium plus supplements. The CyQuant GR dye was used to detect the lysed colonies and the proportional fluorescence to the number and size of colonies was read using a PerkinElmer's VICTOR X5 multilabel plate reader with a 485/535 filter set and 1 s of measurement time. The data are expressed in relative fluorescence units (RFU).

**In vivo xenograft.** Athymic nude female mice (Charles River, Inc (USA), strain Crl:NU(NCr)-Fox1nu) were subcutaneously injected at 7–8 weeks of age in one flank with a total of $10 \times 10^6$ MCF10A, $7 \times 10^6$ A673, and $5 \times 10^5$ Hs578T cells from clones expressing either scrambled or *RPSAP52*-shRNAs, soaked in 100 μl of Matrigel (BD Biosciences). Tumor growth was monitored every 7 days for MCF10A, 3–4 days for A673, and 4 days for Hs578T by measuring tumor width (W, mm) and length (L, mm) until mice were killed at the indicated days post-injection. After allowing to grow for several weeks, tumor volume (V, mm$^3$) was estimated from the formula $V = \frac{\pi \times L \times W^2}{6}$ and tumor weight (g) measured. Animal tests complied with ethical regulations. All the mouse experiments were approved by IDIBELL's Committee for Animal Experimentation.

**Absolute quantification.** Estimations of the absolute amounts of RNAs were obtained by comparison with in vitro transcribed RNA standards of known concentration. These RNA standards correspond to the sequences amplified in RT-qPCR in the analysis of *RPSAP52 + altex*, *RPSAP52 – altex*, *HMGA2* and *LIN28B* transcript expression in all figures, and were generated by introducing the T7 RNA Polymerase promoter upstream of the amplicon by PCR and subsequent in vitro transcription. Serial dilutions of the synthesized RNA standards were used as spike-ins in total RNA extractions from MCF7 cells (which do not express any of the transcripts of interest) and processed in parallel in RT-qPCR with RNA extractions from a known number of MCF10A or A673 cells, so that transcript copy number per cell could be measured. Similarly, for determination of let-7 copy number, we generated a standard curve using synthetic let-7a, b and e purified RNA oligonucleotides (Sigma), corresponding to the sequences hsa-let-7a-5p (5′-rUrGrArGr-GrUrArGrUrArGrGrUrUrGrUrArUrArGrUrU-3′), hsa-let-7b-5p (5′-r UrGrArGrGrUrArGrUrArGrGrUrUrGrUrGrGrUrU-3′), and hsa-let-7e-5p (5′-rUrGrArGrGrUrArGrGrArGrGrUrUrGrUrArUrArGrUrU-3′). For protein quantification, total extracts from a recorded number of cells was analyzed by western blot in parallel with known amounts of the following recombinant proteins: LIN28B (ab134596, Abcam), IGF2BP2 (ab153107, Abcam), HNRNPQ (ab153089, Abcam). Western blot was performed with the following antibodies: anti-LIN28B (ab71415, Abcam, 1:1000), anti-IGF2BP2 (H00010644-M01, Abnova, 1:500), anti-HNRNPQ (NBP1-57197, Novus Biologicals, 1:1000). Band intensity was measured by densitometry with an iBright™ CL1000 Imaging System (ThermoFisher).

**iCLIP-seq.** iCLIP-seq was performed on stable A673 clones. Approximately $8 \times 10^6$ A673 cells stably expressing scrambled shRNA (scr) or shRNA-4 against *RPSAP52* (clone B11) were crosslinked with 150 mJ cm$^{-2}$ total 254-nm irradiation in a Stratalinker 2400. The same amount of non-crosslinked cells were used as controls. Cell lysates were treated with different concentrations (2 or 0.4 U μl$^{-1}$) of RNaseI

(#AM2294, ThermoFisher) and 4 U of Turbo DNase (#AM2238, ThermoFisher) in a final volume of 1 ml. Lysates were then cleared and immunoprecipitated overnight at 4 °C with 10 μg of anti-IGF2BP2 antibody (#RN008P, MBL) preincubated for 1 h at room temperature with 60 μl anti-rabbit IgG Dynabeads (#11204D, ThermoFisher). After two washes in high-salt buffer (50 mM Tris-HCl pH 4.4, 1 M NaCl, 1 mM EDTA, 1% Igepal CA-630, 01% SDS, 0.5% sodium deoxycholate) and one wash in PNK buffer (20 mM Tris-HCl pH 7.4, 10 mM MgCl₂, 0.2% Tween-20), RNA 3′end was dephosphorylated with PNK for 20 min at 37 °C. Beads were then washed once with PNK buffer, once with high-salt buffer and twice with PNK buffer. L3 adapter was then ligated overnight at 16 °C in a 20 μl reaction containing 1.5 μM pre-adenylated L3-App adapter (rAppAGATCGGAAGAGCGGTTCAG// ddC/), 4 μl PEG400 and 10 U T4 RNA Ligase1 (#M0204, New England Biolabs). Beads were then washed twice with high-salt buffer and twice with PNK buffer, and 20% of beads were radioactively labeled with γ-[$^{32}$P]-ATP and 0.5 U μl$^{-1}$ PNK (#M0201, New England Biolabs) for 5 min at 37 °C, added to the remaining cold beads and incubated in 20 μl 1xNuPAGE buffer for 5 min at 70 °C prior to loading the supernatant on a 4–12% NuPAGE Bis-Tris gel (#NP0341BOX, ThermoFisher). The gel was run at 180 V for 50 min, and the protein–RNA complexes were transferred to a nitrocellulose membrane at 30 V for 1 h. The membrane was then autoradiographed through exposure to a film at −80 °C for 1 h. Regions of interest containing the IGF2BP2-RNA crosslinked products were cut out of the membrane and the RNA fragments isolated, reverse transcribed, purified and circularized by incubating with CircLigase II (Epicentre) for 1 h at 60 °C, followed by annealing to Cut_oligo (5′-GTTCAGGATCCACGACGCTCTTCAAAA-3′) and digestion with BamHI. iCLIP libraries were amplified for 27 cycles with P3/P5 Solexa primers, and the appropriate size of products were confirmed by gel electrophoresis. Sequencing of the libraries was performed on a MiSeq instrument following standard manufacturer's procedures, using MiSeq Reagent Kit v3 reagents with single-read, 151-bases read profile. Two independent experiments were performed for each condition. The four libraries were sequenced in two separated pools (#31 & #38) and data acquired with MiSeq Reporter v2.6.3.2. Raw data can be downloaded from https://www.ncbi.nlm.nih.gov/sra/?term=PRJNA484688.

**iCLIP computational analysis.** Read quality was assessed using FastQC (v0.11.7) software (available online at http://www.bioinformatics.babraham.ac.uk/projects/fastqc). After sequencing, PhiX sequences were removed using BWA (Burrows-Wheeler Aligner, v0.7.17)[67]. All the pre-processing steps, peak calling, CITS calling, and annotations were performed using CTK (CLIP Tool Kit) (v1.0.9) software[68], following the recommendations found in https://zhanglab.c2b2.columbia.edu/index.php/ICLIP_data_analysis_using_CTK. After pre-processing steps, final tags from the two biological replicates of the same condition were merged to proceed with peak and CITS calling steps. Peak calling was statistically assessed using a Bonferroni adjusted P-value < 0.05 as a significance threshold. For CITS calling, all tags presenting substitutions were excluded from the analysis, since the abnormally high frequency of substitutions observed could be due to reverse transcriptase read-through. CITS were considered significant with a P-value < 0.001. No proximity clustering was applied in either of the analysis. Adapter trimming, sequence alignments, and alignment manipulations were performed using Cutadapt (v1.16), BWA and samtools (v1.8), respectively. All genome alignments and annotations used hg19 human genome (GCA_000001405.1) as a reference. De novo motif discovery from CITS was performed using HOMER (v4.10) software[69], for which a window of CITS +/−10 nucleotides was taken. Additional python scripts were used for specific CAUH motif enrichment analysis. Genome Browser images were generated using Golden Helix GenomeBrowse® v3.0.0 software (available from http://www.goldenhelix.com). Let-7a/b/e predicted target genes, used for let-7 enrichment analysis in the obtained peaks, were downloaded from miRDB (http://www.mirdb.org/). For differential binding analysis, 3′UTR tags obtained after pre-processing steps by CTK software were used to generate 3′UTR tag counts for each biological replicate of the two conditions. Using these counts, differential binding analysis was performed using DESeq2 bioconductor package v1.18.1[70]. Gene enrichment analyses from significant peaks were conducted using Enrichr v2.0 software (http://amp.pharm.mssm.edu/Enrichr/)[71,72]. Additional graphics and statistical analysis were performed using R v3.4.3 programming language (https://www.R-project.org/).

**Primary tumors expression and methylation analysis.** RNA expression and DNA methylation data from different tumor types was collected from The Cancer Genome Atlas (TCGA) Data Portal (https://tcga-data.nci.nih.gov/tcga). TCGA data were downloaded using TCGAbiolinks v2.9.2[73]. Bioconductor package from the current GDC (Genomic Data Commons) harmonized database aligned against hg38 genome. Box plots analysis represent normalized expression and methylation values corresponding to TCGA COAD, LUSC, LUAD, THCA, and BRCA projects from the GDC data portal (https://portal.gdc.cancer.gov/). We use the Wilcoxon signed-rank test to compare differences between groups. The association between HMGA2 and RPSAP52 expression in primary tumors and cell lines was estimated with a Pearson's correlation. All the statistical analysis and graphical representations were performed using R v3.4.3. For primary samples, an average of HMGA2/RPSAP52 promoter methylation >0.26 (median of the population) was considered as hypermethylated. NCI60 cell lines expression data were downloaded from cBioPortal database (http://www.cbioportal.org/).

**Polysome profile analysis.** MCF10A cells or A673 stable clones were plated in 150 mm dishes and treated with 100 μg ml$^{-1}$ cycloheximide (CHX) at 37 °C for 5 min. Cells were washed twice with cold PBS supplemented with CHX, pelleted and resuspended in 250 μl of hypotonic lysis buffer (1.5 mM KCl, 2.5 mM MgCl₂, 5 mM Tris-HCl pH 7.4, 1 mM DTT, 1% sodium deoxycholate, 1% Triton X-100, 100 μg ml$^{-1}$ CHX) supplemented with mammalian protease inhibitors (SIGMA) and RNase inhibitor (NEB) at a concentration of 100 U ml$^{-1}$, and left on ice for 5 min. Cell lysates were cleared of debris and nuclei by centrifugation for 5 min at 17,000 × g. Protein concentrations were determined by BCA assay and 500 μg of lysate were loaded on 10–50% sucrose linear gradients containing 80 mM NaCl, 5 mM MgCl₂, 20 mM Tris-HCl pH 7.4, 1 mM DTT, 10 U ml$^{-1}$ RNase inhibitor with a BIOCOMP gradient master. Gradients were centrifuged on a SW40 rotor for 3.5 h at 217,290 × g. Gradients were analyzed on a BIOCOMP gradient station and collected in 11 (MCF10A) or 13 (A673) fractions ranging from light to heavy sucrose. Fractions were supplemented with SDS at a final concentration of 1% and placed for 10 min at 65 °C. To each fraction was added 1 ng of firefly luciferase mRNA, followed by phenol–chloroform extraction and precipitation with iso-propanol. Purified RNAs from each fraction were retrotranscribed and subjected to qPCR. mRNA quantification was normalized to firefly mRNA.

**IGF2BP2 coimmunoprecipitation and mass spectrometry analysis.** Immuno-precipitation and sample digestion for mass spectrometry analysis: 1 mg of pre-cleared protein extract from three replicates of control cells (scr) and cells depleted for RPSAP52 (sh4 B11 clone) were immunoprecipitated overnight at 4 °C using 5 μg of anti-IGF2BP2 antibody (#H00010644-M01, Abnova) and 40 μl of Dyna-beads® M-280 anti-mouse IgG beads (#11202D, ThermoFisher) in 1 ml RIP buffer (150 mM KCl, 25 mM Hepes at pH 7.9, 5 mM EDTA, 0.5 mM DTT, 0.5% NP40, 1× protease inhibitor cocktail (Roche)). After three washes with RIP buffer, the resulting material was in-bead digested with trypsin. Briefly, the beads were washed three times with 500 ml of 200 mM Ammonium Bicarbonate (ABC) and resus-pended in 60 ml of 6 M Urea 200 mM$^{-1}$ ABC. The samples were then reduced with 10 ml of 10 mM DTT (1 h, 30 °C) and alkylated with 10 ml of 20 mM Iodoaceta-mide (30 min, room temperature and darkness). After that, the samples were diluted with 280 ml of 200 mM ABC and digested with 5 ml of 0.2 mg ml$^{-1}$ Trypsin for 16 h at 37 °C. The beads were finally pulled down (5 min at 5000 g), the supernatant transferred to new, cleaned tubes and acidified with 20 ml of 100% Formic acid. The resulting peptides mixtures were desalted using C18 stage tips (UltraMicroSpin Column, The Nest Group, Inc., MA) and dried in a SpeedVac.

Mass spectrometry analysis (LC-MS/MS): the dried-down peptide mixtures were analyzed in a nanoAcquity liquid chromatography (Waters) coupled to a LTQ-OrbitrapVelos (ThermoScientific) mass spectrometer. The tryptic digests were resuspended in 10 μl 1% FA solution and an aliquot of 3 μl of each sample was injected for chromatographic separation. Peptides were trapped on a Symmetry C18TM trap column (5 μm, 180 μm × 20 mm; Waters), and were separated using a C18 reverse phase capillary column (ACQUITY UPLC BEH column; 130 Å, 1.7 μm, 75 μm × 250 mm, Waters). Eluted peptides were subjected to electrospray ionization in an emitter needle (PicoTipTM, New Objective) with an applied voltage of 2000 V. Peptide masses (300–1700 m/z) were analyzed in data dependent mode, where a full scan MS was acquired in the Orbitrap with a resolution of 60,000 FWHM at 400 m/z. Up to the 15th most abundant peptides (minimum intensity of 500 counts) were selected from each MS scan and then fragmented in the linear ion trap using CID (38% normalized collision energy) with helium as the collision gas. The scan time settings were Full MS: 250 ms (1 microscan) and MSn: 120 ms. Generated .raw data files were collected with ThermoXcalibur (v2.2).

Data analysis: the .raw files were analyzed with the MaxQuant(v.1.6.2.6a) software using the built-in search engine Andromeda to search against the Swissprot Human database downloaded from UniprotKB website in March 26, 2018. The search parameters were set as follow: the enzyme was trypsin with a maximum of two allowed missed cleavages. Oxidation in methionines as well as Acetylation at protein N-terminal was set as variable modifications while carbamidomethylation in cysteines was set as fixed modification. The mass tolerances for the first and main search were set at 20 and 4.5 ppm, respectively. In addition, only peptides with more than 6 and up to 25 aminoacids were considered. The final list of identified peptides and proteins were filtered by using a 5% false discovery rate (FDR) both at peptide and protein level. To enhance the identification of proteins the match between runs option was selected.

Protein–protein interaction analysis: the statistical analysis of the protein–protein interactions found in our experimental conditions was performed with the help of the Significance Analysis of the INTeractome (SAINT) algorithm which was implemented in the http://statsms.crg.es/ site.

**Expression arrays.** Total RNA from each sample was quantified using the NanoDrop ND-1000 and RNA integrity was assessed by standard denaturing agarose gel electrophoresis. For microarray analysis, Agilent Array platform was employed. The sample preparation and microarray hybridization were performed based on the manufacturer's standard protocols. Briefly, total RNA from each sample was amplified and transcribed into fluorescent cRNA with using the manufacturer's Agilent's Quick Amp Labeling protocol (version 5.7, Agilent Technologies). The labeled cRNAs were hybridized onto the Whole Human Genome Oligo Microarray (4 × 44 K, Agilent Technologies). After having washed

the slides, the arrays were scanned by the Agilent Scanner G2505C. Agilent Feature Extraction software (version 11.0.1.1) was used to analyze acquired array images. Quantile normalization and subsequent data processing were performed using the GeneSpring GX v12.1 software (Agilent Technologies). Differentially expressed genes were identified through Fold Change filtering and Volcano filtering. Pathway analysis and GO Analysis were applied to determine the roles of these differentially expressed genes.

**Statistical analysis.** Bar graphics and statistical comparisons were obtained with the GraphPad Prism 8.1.2 software. Comparative analyses between different experimental groups were performed using $t$-student test and one-way ANOVA with Bonferroni's or Dunnet's post hoc tests for intergroup comparisons. For cancer patients' samples, we used the Kaplan–Meier method for survival analysis and the log-rank test was used to analyze the differences between the groups. Cox regression method was used to analyze the independent prognostic importance of expression or methylation. Results of the univariate Cox regression analysis are represented by the hazards ratio (HR) and 95% confidence interval (CI). Results were considered significant if the $P$-value was <0.05 (*), <0.01 (**), <0.001 (***), or <0.0001 (****). Unless otherwise stated, data are presented as the mean ±SD.

**Unprocessed scans.** All unprocessed and uncropped scans and images can be found in the source data file.

## Data availability

All relevant data are available from the authors. Raw data for the iCLIP-seq experiment have been deposited under the accession code PRJNA484688 (https://www.ncbi.nlm.nih.gov/sra/?term=PRJNA484688). iCLIP-seq peaks and CITS are provided in the Oliveira-Mateos et al_Supplementary Data 1. The list of significant altered transcripts from the microarray expression analysis is provided in the Oliveira-Mateos et al_Supplementary Data 2. Numerical source data for Figs. 1e–g, 2c, 3a–c, e–g, 4b, d, f, 5a, c, 6a–e, 7b; Supplementary Figs. 1c, 2b–e, 3c, d, f–h, d–f, 5g, j, 6a, c and 7b and all unprocessed images and scans for Figs. 1d, f, 2a, b, d, f, g, 3d, f, g, 4b, c, e, 5a–c, 6f; Supplementary Figs. 2a, 3e–h, 4d, 5a, e, h, i, and 6b can found in the source data file.

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

## Acknowledgements

We thank CERCA Program/Generalitat de Catalunya for their institutional support. This work was also supported by the Fundació La Marató de TV3, grant number #20131610 (S.G.), the AECC-Junta de Barcelona (S.G.), the *Fundación Científica de la AECC* under grant GCB13131578DEÁ (O.M.T.), the Ministerio de Economía y Competitividad (MINECO) and the Instituto de Salud Carlos III (ISCIII), co-financed by the European Development Regional Fund, 'A way to achieve Europe' ERDF, under grant numbers SAF2014-56894-R (S.G.), SAF2014-55000-R (M.E.), CB16/12/00312 (M.E.) and PI16/01898 (A.V.), the Health and Science Departments of the Catalan Government (Generalitat de Catalunya), and the European Union's Horizon 2020 research and innovation program under the Marie Sklodowska-Curie, grant agreement number 799850 (A.V., L. F.). C.O.-M. is a pre-doctoral fellow funded by the Basque Government (PRE_2013_1_1009). A.O.-G. is a pre-doctoral fellow funded by MINECO (BES-2015-071452). M.E. is an ICREA Research Professor.

## Author contributions

Conception and design: S.G., C.O.-M. and L.F. Development of methodology: C.O.-M., A.S.-C., A.O.-G., R.B.-S., M.S., T.R., J.P., A.G., M.M.-I., D.H.-M., L.F. and S.G. Acquisition of data (provided animals, acquired and managed patients, provided facilities, etc): O.M.T., A.V., and M.E. Analysis and interpretation of data (e.g., statistical analysis, biostatistics, computational analysis): D.P., M.E.C.-C., M.C.deM., A.M.-C. and A.G. Writing, review and/or revision of the paper: All authors. Administrative, technical or material support (i.e., reporting or organizing data, constructing databases): D.P., M.E. C.-C. and M.C.deM. Study supervision: S.G.

## Additional information

**Competing interests:** A. Villanueva is founder and shareholder of Xenopat S.L. (Barcelona, Spain). The other authors declare no competing interests.

