## [Peer Review File · Nature Communications]

Reviewers' comments:

Reviewer #1, Expertise: IGFBP pathways (Remarks to the Author):

The work by Oliveira-Mateos et al. describes the effect of the transcribed pseudogene RPSAP52 on the function of HMGA2 and IGF2BP2 as well as on let-7 miRNA expression. The work is an extension of the authors' earlier observations that RPSAP52, which runs antisense to HMGA2, positively regulates HMGA2 expression through the formation of an R loop structure. The investigators now address the functional role of RPSAP52 in breast carcinoma and sarcoma cell lines and uncover its implication in the regulation of IGF2BP2 function and let-7 expression.

Using RNA pulldown combined with mass spectrometry the investigators identify IGF2BP2 as a RPSAP52 protein partner. They then show that RPSAP52 depletion leads to a decrease in IGF2BP2 and RAS protein levels by a mechanism that may implicate let-7. They show that RPSAP52 depletion leads to decreased MCF10A cell proliferation, colony formation and migration as well as tumorigenicity in vivo.

Upon addressing the effects of RPSAP52 depletion in the Ewing sarcoma cell line A673, the investigators observe that Lin28B expression decreases but that IGF2BP2 expression is unaffected, whereas expression of let-7 miRNA family members increases. By performing iCLIP, they show that the function of IGF2BP2 appears to be at least partially impaired by RPSAP52 depletion as recognition of consensus sequences in the 3'UTR of cadet target genes diminishes. Several of the target genes are implicated in the cell cycle and various functions exploited by malignant cells.

The investigators conclude that RPSAP52 controls the HMGA2/IGF2BP2/LIN28B axis through a mechanism that includes transcriptional regulation of HMGA2 and the regulation of IGF2BP2 function on its downstream targets, which among other effects, results in decreased let-7 expression and derepression of their potential oncogenic targets.

This is a potentially interesting study that highlights the implication of a lncRNA in tumor promotion by affecting the lin28B/IGF2BP2/Let-7 axis. However, it requires clarification and some additional experiments, as the conclusions remain fairly speculative, particularly regarding the mechanism by which RPSAP52 exerts its effects.

Some key issues:

1. Choice of cell lines and tumorigenicity assays:

The investigators use the MCF10A cell lines for their tumorigenicity assays. This appears to be a somewhat curious choice as MCF10A cells typically represent normal breast and are non-tumorigenic. Are the investigators' cells transformed? Have they become tumorigenic spontaneously? It would be useful to repeat these experiments in a bona fide breast cancer cell line.

Having repeated several of the experiments in A673 cells, it seems surprising that the investigators did not assess the effect of RPSAP52 depletion on tumor formation by these cells, particularly given the distinct effects that the depletion appears to have on IGF2BP2 expression in the two cell lines. Given the the conclusions that the investigators draw, the effect of RPSAP52 on the tumorigenicity of A673 cells is important to determine whether it is broadly applicable in different cellular contexts or not.

2. The functional effect of RPSAP52 on the Lin28B/IGF2BP2/Let-7 axis:

Whereas the observations by the investigators are clearly of potential interest, they require clarification as the effect of RPSAP52 appears to be different in Lin28B+ and Lin28B- cells. For instance, IGF2BP2 is down regulated upon RPSAP52 depletion in Lin28B- MCF10A cells but not in Lin28B+ A673 cells, in which its function appears to be altered. Is this difference due to the

presence or absence of Lin28B or some other property intrinsic to the cells? It would be necessary to reconstitute MCF10A cells with Lin28B and conversely deplete A673 cells of Lin28B and then assess the effect of removing RPSAP52 on IGF2BP2 expression and function.

3. Mechanism of action:

There is much speculation as to how RPSAP52 exerts its effects. The investigators suggest that the lncRNA may alter the composition of the RNP particles that contain IGF2BP2. This is indeed a possibility but it needs to be substantiated.

4. Stemness:

The investigators state that RPSAP52 plays a role in stemness maintenance based on the observation that its depletion leads to down regulation of OCT4 and NANOG. However, no functional assays are conducted to support this notion. The investigators should at the very least conduct clonogenicity assays to verify their claim.

In summary, this is a study of potential interest to the field but it is still rather preliminary and requires more in-depth assessment of the mechanisms by which RPSAP52 exerts its function in addition to addressing the generality of its effect on tumor promotion.

Reviewer #2, Expertise: lncRNA (Remarks to the Author):

This manuscript describes multiple studies characterizing the molecular and biological effects of a pseudogene lncRNA, RPSAP52, on cell proliferation and tumorigenesis. The authors show that RPSAP52 is abundantly expressed in the cytoplasm where it associates with IGF2BP2/IMP2 and HNRNP Q. Downregulation alters the relationship between LIN28B and let-7 miRNAs. Finally, high levels of the lncRNA are associated with cancers. Overall, while this work describes yet another lncRNA that is involved in cancer, I found the quality of the experimental approaches to be appropriate and sound, though leading to mostly suggestive and not strongly compelling conclusions at this point.

While not criticizing most molecular approaches, I do have some significant problems with the work as presented. In my opinion, these must be addressed.

1. A major concern is that we still do not know some important and relevant facts about this system. RPSAP52 is annotated as a lncRNA and one of several RPSA pseudogenes. As RPSAP52 is cytoplasmic, is it polysome associated? As some annotated lncRNAs have recently been shown to express short proteins or polypeptides, might this be the case for it?
2. What is its actual abundance in cells (molecules per cell) and how is this related to the cellular abundance of IGF2BP2, HNRNP Q and let-7?
3. Altered expression of RPSA itself has been shown to be associated with cancers. As pseudogenes harbor multiple mutations and are expressed from different genomic locations, it is expected that they will be expressed differently in cells and may even bind differently to proteins. The authors rule out effects of other pseudogenes in their knockdown experiments, but curiously never examine RPSA itself. What is the expression level of RPSA in the experiments and might its RNA be affected by the shRNAs and might its mRNA associate strongly with IGF2BP2 and HNRNP Q? How might altered RPSA expression (protein or mRNA) impact the results shown?

Minor comment: The running title is somewhat misleading: it gives the impression that RPSAP52 is an antisense to HMGA2. While expressed from the opposite strand, the RNAs are actually not at all complementary to each other, as intron 1 of RPSAP52 is the complement of HMGA2, but is spliced out. The transcription is antisense, but the transcripts are not.

We thank the reviewers for their constructive comments, which made the revised manuscript much stronger. All major changes in the revised manuscript are marked in blue.

Reviewers' comments:

Reviewer #1, Expertise: IGFBP pathways (Remarks to the Author):

The work by Oliveira-Mateos et al. describes the effect of the transcribed pseudogene RPSAP52 on the function of HMGA2 and IGF2BP2 as well as on let-7 miRNA expression. The work is an extension of the authors' earlier observations that RPSAP52, which runs antisense to HMGA2, positively regulates HMGA2 expression through the formation of an R loop structure. The investigators now address the functional role of RPSAP52 in breast carcinoma and sarcoma cell lines and uncover its implication in the regulation of IGF2BP2 function and let-7 expression.

Using RNA pulldown combined with mass spectrometry the investigators identify IGF2BP2 as a RPSAP52 protein partner. They then show that RPSAP52 depletion leads to a decrease in IGF2BP2 and RAS protein levels by a mechanism that may implicate let-7. They show that RPSAP52 depletion leads to decreased MCF10A cell proliferation, colony formation and migration as well as tumorigenicity in vivo.

Upon addressing the effects of RPSAP52 depletion in the Ewing sarcoma cell line A673, the investigators observe that Lin28B expression decreases but that IGF2BP2 expression is unaffected, whereas expression of let-7 miRNA family members increases. By performing iCLIP, they show that the function of IGF2BP2 appears to be at least partially impaired by RPSAP52 depletion as recognition of consensus sequences in the 3'UTR of cadet target genes diminishes. Several of the target genes are implicated in the cell cycle and various functions exploited by malignant cells.

The investigators conclude that RPSAP52 controls the HMGA2/IGF2BP2/LIN28B axis through a mechanism that includes transcriptional regulation of HMGA2 and the regulation of IGF2BP2 function on its downstream targets, which among other effects, results in decreased let-7 expression and derepression of their potential oncogenic targets.

This is a potentially interesting study that highlights the implication of a lncRNA in tumor promotion by affecting the lin28B/IGF2BP2/Let-7 axis. However, it requires clarification and some additional experiments, as the conclusions remain fairly speculative, particularly regarding the mechanism by which RPSAP52 exerts its effects.

We are grateful for the reviewer's positive comments. Following his/her suggestions, we believe we now provide key mechanistic details about how *RPSAP52* does indeed

regulate the oncogenic axis. We have also widened the scope of our experimental model system by including 2 new breast cancer cell lines and by introducing new *in vivo* experiments that have strengthened our conclusions.

Some key issues:

1. Choice of cell lines and tumorigenicity assays:

The investigators use the MCF10A cell lines for their tumorigenicity assays. This appears to be a somewhat curious choice as MCF10A cells typically represent normal breast and are non-tumorigenic. Are the investigators' cells transformed? Have they become tumorigenic spontaneously?

It would be useful to repeat these experiments in a bona fide breast cancer cell line.

Although our observations indicate a relevant role for *RPSAP52* in the non-transformed MCF10A cell line and as such, the data is useful for revealing the potential physiological role of this pseudogene in non-tumorigenic settings, the reviewer is right to point out that other breast cancer cell lines needed to be tested to support the role in oncogenesis. We have repeated *in vitro* and *in vivo* experiments with Hs578T and HCC1143 cell lines (which are among the cell lines with highest expression of *RPSAP52*, Fig. 1e) to confirm the tumorigenic features of the noncoding transcript at the phenotypic (new Figures 3a, 3b, 3g) and molecular levels (new Supplementary Figures 3g and h).

Having repeated several of the experiments in A673 cells, it seems surprising that the investigators did not assess the effect of *RPSAP52* depletion on tumor formation by these cells, particularly given the distinct effects that the depletion appears to have on *IGF2BP2* expression in the two cell lines. Given the the conclusions that the investigators draw, the effect of *RPSAP52* on the tumorigenicity of A673 cells is important to determine whether it is broadly applicable in different cellular contexts or not.

Thanks for the suggestion. We have now tested the impact of depleting *RPSAP52* in A673 *in vivo* (new Figure 4f, showing subcutaneous injections in mice). We would like to point out that this has been done with the shRNA clone that downregulates *RPSAP52* but not *HMGA2* mRNA levels (sh3, as shown in Supplementary Figure 7b). The reduction in tumor growth that we observe, together with the survival curves in patients (Figure 7c, where high expression of *RPSAP52*, but not of *HMGA2*, indicates worse outcome) reinforce the tumorigenic features of *RPSAP52* also in the context of sarcoma.

2. The functional effect of *RPSAP52* on the Lin28B/*IGF2BP2*/*Let-7* axis:

Whereas the observations by the investigators are clearly of potential interest, they require clarification as the effect of *RPSAP52* appears to be different in Lin28B+ and Lin28B- cells. For instance, *IGF2BP2* is down regulated upon *RPSAP52* depletion in

Lin28B- MCF10A cells but not in Lin28B+ A673 cells, in which its function appears to be altered. Is this difference due to the presence or absence of Lin28B or some other property intrinsic to the cells? It would be necessary to reconstitute MCF10A cells with Lin28B and conversely deplete A673 cells of Lin28B and then assess the effect of removing RPSAP52 on IGF2BP2 expression and function.

These are insightful suggestions and we have followed the reviewer's advice in the revision. On one hand, and in contrast to the situation in A673, we have confirmed that LIN28B is generally expressed at very low levels in breast cancer cell lines (new Supplementary Figure 4e). Similarly to what happened in MCF10A, depletion of *RPSAP52* in Hs578T and HCC1143 cell lines resulted in a marked decrease in IGF2BP2 protein levels (new Supplementary Figure 3g and 3h), confirming the generality of the previous observations and indicating an effect independent of LIN28B action. Interestingly, even though overexpression of LIN28B in MCF10A cells counteracts the effect of *RPSAP52* depletion (new Figure 2g), depletion of LIN28B in A673 cells does not impact on IGF2BP2 levels or function (new Supplementary Figure 5h-j). Altogether, these data suggest independent ways of action for *RPSAP52* and LIN28B on IGF2BP2, and a specific stability of IGF2BP2 protein in A673 cells for which we have no definite explanation as of yet. One possible explanation is that the levels of *RPSAP52* are ~1 order of magnitude lower in A673 cells than in breast cancer cells, while IGF2BP2 protein levels are similar between cell lines (new Supplementary Figure 4d).

3. Mechanism of action:

There is much speculation as to how RPSAP52 exerts its effects. The investigators suggest that the lncRNA may alter the composition of the RNP particles that contain IGF2BP2. This is indeed a possibility but it needs to be substantiated.

This is indeed an important point and we have now addressed the mechanistic aspects of *RPSAP52* function more in more detail. Co-immunoprecipitation experiments coupled to mass spectrometry analysis in the background of +/- *RPSAP52* did not reveal any changes in the affinity of IGF2BP2 protein for its binding partners (new Supplementary Figure 6b, 6c). However, given the role of IGF2BP2 as a co-translational regulator and the changes in binding to a subset of 3'UTRs revealed by the iCLIP-seq experiment, we have analyzed the distribution of IGF2BP2 and its key mRNA targets across sucrose gradients (new Figure 6). Importantly, the data indicates a loss of IGF2BP2 recruitment on large polysomes upon *RPSAP52* depletion, concomitant with a decrease in *HMGGA2* and *LIN28B* mRNA co-sedimentation in polysomes. In accordance with iCLIP data, other IGF2BP2 targets, such as *NRAS* mRNA, are not altered in the gradients. We think this is strong evidence that supports the role of *RPSAP52* as an important agent in the translational regulation of specific mRNA targets through IGF2BP2 recruitment to polysomes.

4. Stemness:

The investigators state that RPSAP52 plays a role in stemness maintenance based on the observation that its depletion leads to down regulation of OCT4 and NANOG. However, no functional assays are conducted to support this notion. The investigators should at the very least conduct clonogenicity assays to verify their claim.

We are now providing a clonogenicity assay on soft agar (new Figure 3e) to strengthen this point. We thank the reviewer for the suggestion.

In summary, this is a study of potential interest to the field but it is still rather preliminary and requires more in-depth assessment of the mechanisms by which RPSAP52 exerts its function in addition to addressing the generality of its effect on tumor promotion.

We are grateful for the reviewer's positive comments and suggestions about our work. We believe the regulatory model we propose is now strengthened in both the phenotypic and the mechanistic aspects.

Reviewer #2, Expertise: lncRNA (Remarks to the Author):

This manuscript describes multiple studies characterizing the molecular and biological effects of a pseudogene lncRNA, RPSAP52, on cell proliferation and tumorigenesis. The authors show that RPSAP52 is abundantly expressed in the cytoplasm where it associates with IGF2BP2/IMP2 and HNRNP Q. Downregulation alters the relationship between LIN28B and let-7 miRNAs. Finally, high levels of the lncRNA are associated with cancers. Overall, while this work describes yet another lncRNA that is involved in cancer, I found the quality of the experimental approaches to be appropriate and sound, though leading to mostly suggestive and not strongly compelling conclusions at this point.

We thank the reviewer for his/her general positive appreciation of our work. Following his/her suggestions, we have addressed the points raised concerning a more detailed characterization of the processed pseudogene itself and of the parental *RPSA* gene. Also, the additional mechanistic data that we are providing in the revised manuscript clarifies the role of *RPSAP52* as a regulator of IGF2BP2 recruitment to polysomes and the subsequent regulation of the translation of key target mRNAs.

While not criticizing most molecular approaches, I do have some significant problems with the work as presented. In my opinion, these must be addressed.

1. A major concern is that we still do not know some important and relevant facts about this system. RPSAP52 is annotated as a lncRNA and one of several RPSA pseudogenes.

As *RPSAP52* is cytoplasmic, is it polysome associated? As some annotated lncRNAs have recently been shown to express short proteins or polypeptides, might this be the case for it?

The reviewer raises an important point and we now provide new experiments to clarify this issue. On one hand, *in vitro* translation assays in reticulocyte lysates do not support the notion that *RPSAP52* is translated (new Supplementary Figure 2a). On the other hand, analysis of sucrose gradients indicates that both *RPSAP52* isoforms are associated with polysomes in both MCF10A and A673 cells (new Supplementary Figure 2b-e and Supplementary Figure 6a). Critically, rather than reflecting translation of *RPSAP52* itself, changes in the recruitment of IGF2BP2 protein and of *HMGA2* and *LIN28B* mRNAs to polysomes upon *RPSAP52* depletion suggest the possibility that the pseudogene is involved in translational regulation of other mRNAs by facilitating the distribution of IGF2BP2 in large polysomes (new Figure 6a-f). This new evidence strengthens the mechanistic part of the manuscript and validates our proposed model, that *RPSAP52* acts mainly through modulation of IGF2BP2 function.

2. What is its actual abundance in cells (molecules per cell) and how is this related to the cellular abundance of IGF2BP2, HNRNP Q and *let-7*?

The reviewer addresses a relevant point, which is the actual number of transcripts corresponding to *RPSAP52* that exist in our samples. We have carried out absolute quantifications of *let-7* miRNAs, *RPSAP52* transcripts and *HMGA2* and *LIN28B* mRNAs both in MCF10A and A673 cells. We have used a direct qPCR comparison, in which a standard curve was built with increasing amounts of *in vitro* transcribed templates corresponding to the same amplicons that result from RT-qPCR with cellular RNAs (new Supplementary Figure 4d and Supplementary Methods). This approach has revealed that the number of total *RPSAP52* transcripts in MCF10A cell line is ~300 molecules/cell, and in A673 ~30 molecules/cell (for *HMGA2* mRNA, it is ~1750 and ~1000, respectively), which is consistent with a regulatory cytoplasmic role and confirms the difference in ~1-2 orders of magnitude between the expression of the coding and the noncoding transcript we had observed in patients (Supplementary Figure 4c). Interestingly, *let-7a*, *b* and *e* together are present at ~300 copies/cell in MCF10A and ~90 copies per cell in A673. *RPSAP52* is thus nearly as abundant as these well known regulatory noncoding RNAs. Additionally, *LIN28B* mRNA was estimated in A673 cells at ~150 molecules/cell, which is stoichiometrically comparable to *RPSAP52* and fits well with their presence in common complexes.

Additionally, we have also performed the estimation of IGF2BP2, HNRNPQ and *LIN28B* protein levels (new Supplementary Figure 4d and Supplementary Methods). According to our data, they are present at comparable levels but at ~4 orders of magnitude higher than the mentioned RNAs, suggesting that subcellular localization (e.g., IGF2BP2 protein is abundantly present in the nucleus, see picture below) and local concentration may be highly relevant for the formation of RNA:protein binary or ternary complexes.

Immunofluorescence staining of IGF2BP2 in A673 cells. (In red, anti-IGF2BP2 antibody: #RN008P, MBL; in blue, DAPI).

3. Altered expression of RPSA itself has been shown to be associated with cancers. As pseudogenes harbor multiple mutations and are expressed from different genomic locations, it is expected that they will be expressed differently in cells and may even bind differently to proteins. The authors rule out effects of other pseudogenes in their knockdown experiments, but curiously never examine RPSA itself. What is the expression level of RPSA in the experiments and might its RNA be affected by the shRNAs and might its mRNA associate strongly with IGF2BP2 and HNRNP Q? How might altered RPSA expression (protein or mRNA) impact the results shown?

This is an important point and we have now analyzed the levels of RPSA in our experimental models, including the new Hs578T cell line for which additional *in vivo* and *in vitro* experiments are provided. In all cases, RPSA levels remained essentially unchanged upon *RPSAP52* knockdown (new Supplementary Figure 3e). Also, iCLIP-seq experiments did not reveal binding of IGF2BP2 protein to *RPSA* mRNA, nor did both proteins interact in our co-IP mass-spec analysis (new Supplementary Figure 6c). Altogether, we do not think RPSA protein is involved in the regulatory network we are describing.

Minor comment: The running title is somewhat misleading: it gives the impression that *RPSAP52* is an antisense to *HMGA2*. While expressed from the opposite strand, the RNAs are actually not at all complementary to each other, as intron 1 of *RPSAP52* is the complement of *HMGA2*, but is spliced out. The transcription is antisense, but the transcripts are not.

The reviewer is right in his/her comment and we have now changed the running title to “A processed pseudogene from *HMGA2* locus promotes cell growth through IGF2BP2 pathway” to better describe the organization of the locus.

REVIEWERS' COMMENTS:

Reviewer #1 (Remarks to the Author):

The investigators have addressed the points raised and have substantially improved the manuscript. As indicated in my initial appraisal, the manuscript describes the implication of an lncRNA in tumor promotion through modulation of the Lin28B/IGF2BP2/Let-7 axis. The additional experiments that the investigators have performed have contributed to strengthening the mechanistic basis of their conclusions.

The study will be of interest to researchers in the field and beyond. It contributes significantly to our understanding of the intricacies of how IGF2BP2 functions and to the role of lncRNAs in cancer.

Reviewer #2 (Remarks to the Author):

The authors have nicely addressed all my concerns.

We thank again the reviewers for their very positive comments on the revised version of the manuscript. Their requests and suggestions made the revised manuscript much stronger.

REVIEWERS' COMMENTS:

Reviewer #1 (Remarks to the Author):

The investigators have addressed the points raised and have substantially improved the manuscript. As indicated in my initial appraisal, the manuscript describes the implication of an lncRNA in tumor promotion through modulation of the Lin28B/IGF2BP2/Let-7 axis. The additional experiments that the investigators have performed have contributed to strengthening the mechanistic basis of their conclusions.

The study will be of interest to researchers in the field and beyond. It contributes significantly to our understanding of the intricacies of how IGF2BP2 functions and to the role of lncRNAs in cancer.

We thank very much the reviewer for his/her very favorable appraisal of our work. We agree the revised version has specially strengthened the mechanistic aspects of the manuscript.

Reviewer #2 (Remarks to the Author):

The authors have nicely addressed all my concerns.

We equally thank reviewer #2 for the previous requests made and the positive appraisal of the revision.